# Chain-of-Thought Gradient Descent

**Hong-Yu Chen** [* 1]  **Venkat Sripad Ganti** [* 2]  **Hude Liu**  **Jerry Yao-Chieh Hu** [1]  **Han Liu** [1 3]

## Abstract

We show that Chain-of-Thought (CoT) expands the expressiveness of Transformer in-context learning (ICL). Specifically, we show CoT enable efficient simulation of In-Context Gradient Descent (ICGD) for $N$-layer neural network. Different from CoT, a Transformer with fixed depth and hidden dimension has fixed ICL capacity in one forward pass. Simulating larger models or more optimization steps in-context requires deeper or wider Transformers. CoT removes this limitation by providing an expandable workspace via the sequence trajectory. This enables arbitrary-step and arbitrary-capacity ICGD within a constant-depth Transformer. Second, we provide a provable efficient guarantee unique to CoT through dynamical masking. The attention mechanism only process the relevant tokens for the current update step. This eliminates the redundant "process everything" cost of single-pass deep models. Specifically, we prove this CoT mechanism improves the computational cost of the prior best in-context result [Wu et al., ICML 2025] by $O(N)$. Numerical validations support our theory.

## 1 Introduction

We show that frozen transformer with CoT simulates training dynamic of neural network in-context. Specifically, we prove that single-layer transformer with CoT simulates arbitrary steps of gradient descent for $N$-layer neural network.

Our construction uses dynamic masking to enforce local access to the trajectory, so each step reads only the relevant tokens for current forward/backward update. This eliminate the redundancy that arises when all model parameters and intermediate states are processed at every step. As a result, we obtain $O(N)$ computation improvement over prior deep-transformer ICGD constructions (Wu et al., 2025).

A growing line of work explains in-context learning through in-context gradient descent (ICGD). Given in-context examples, a frozen transformer simulates the multi-step gradient descent of machine learning models (Akyürek et al., 2023; Dai et al., 2023; Cheng et al., 2024; Von Oswald et al., 2023; Bai et al., 2023; Wang et al., 2024; Wu et al., 2025). This theoretical result is also backed up by empirical analysis (Garg et al., 2022). However, existing ICGD constructions face a structural bottleneck when the target model is large (e.g., an $N$-layer network). Since the only place to store the simulated model's parameters is the input, prior works pack all network weights and intermediate state into each token embedding (Wu et al., 2025; Wang et al., 2024). This forces the embedding dimension to scale with $N$ and induces redundant computation. Each transformer layer must process the entire embedding even when the current computation depends only on a small slice of the state (see Figure 1). Furthermore, simulating larger models or more ICGD steps requires increasing the hidden dimension or stacking additional transformer layers.

CoT changes this computational regime. It changes the location of the simulated state. Single-pass deep models compress the entire state into the hidden dimension. In contrast, CoT writes intermediate computations of ICGD as discrete tokens across an expanding sequence trajectory. This externalization provides an expandable workspace. A constant-depth transformer uses this workspace to execute arbitrary step of ICGD for models of arbitrary capacity. This provide theoretical guarantee of CoT mechanism. Furthermore, we establish an efficiency guarantee unique to the CoT setting. The externalized state allows the model to selectively access its history. We formalize this capability through dynamic masking. Dynamic masking restricts the model to process only the relevant tokens for the current update. This mechanism eliminates the redundant "process everything" cost of single-pass deep models. Recent work addresses CoT efficiency through a reduction rule

---

[1]Center for Foundation Models and Generative AI; Department of Computer Science, Northwestern University, Evanston, IL, USA; NSF – Simons AI Institute for the Sky (SkAI), Chicago, USA [2]Department of Mathematics, Northwestern University, Evanston, IL, USA [3]Department of Statistics and Data Science, Northwestern University, Evanston, IL, USA. Correspondence to: Hong-Yu Chen <charlie.chen@u.northwestern.edu>, Venkat Sripad Ganti <venkatganti2027@u.northwestern.edu>, Hude Liu <hudeliu0208@gmail.com>, Jerry Yao-Chieh Hu <jhu@u.northwestern.edu>, Han Liu <han-liu@northwestern.edu>.

*Proceedings of the 43$^{rd}$ International Conference on Machine Learning*, Seoul, South Korea. PMLR 306, 2026. Copyright 2026 by the author(s).

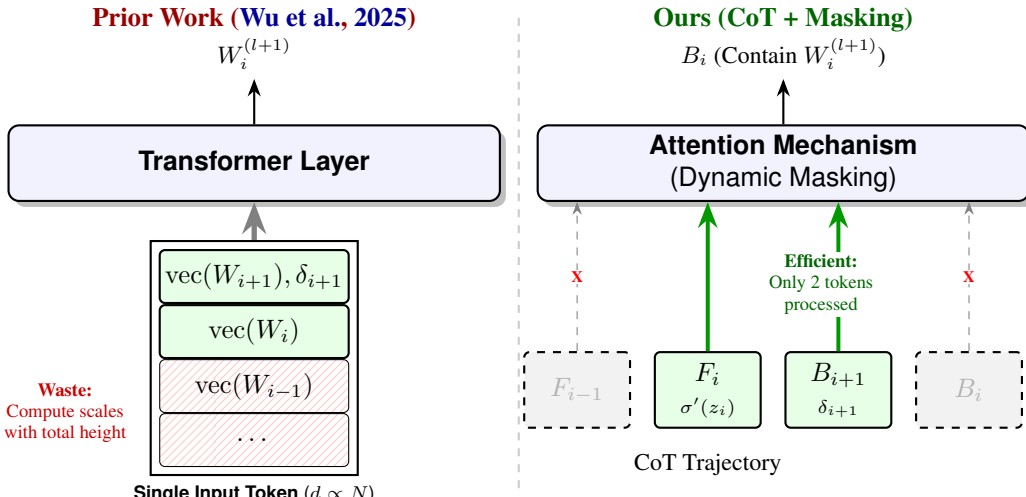

*Figure 1.* **Comparison of Computational Cost.** We demonstrate the concept on one-step backpropagation on $i$-th layer of neural network. **Left:** Wu et al. (2025) stack all weights $\text{vec}(W_{1:N})$ vertically into a single high-dimensional input vector, forcing the transformer to process the full stack every step. **Right:** Our approach keeps weights in separate tokens ($F, B$). Dynamic Masking (Green Arrows) retrieves only the relevant tokens ($F_i, B_{i+1}$), keeping the computation constant $O(1)$. We omit the whole CoT trajectory for simplicity.

that deletes irrelevant tokens (Yang et al., 2025). Our dynamic masking emphasizes selective state exposure rather than permanent token deletion. See Section 5 for detailed discussion.

In Section 3, we demonstrate the intuition of dynamic masking in a simpler optimization settings (e.g., GLMs, ridge, lasso). We show that bounded context access suffices for multi-step approximation with a constant depth transformer and controls per-step computation. In Section 4, we show the same mechanism scales to the training dynamics of deep networks, enabling a single transformer block to simulate the forward/backpropagation update across $N$ layers with improved complexity.

**Contributions.** We summarize our contributions as follow:

- **CoT-Driven Simulation of Deep-Network Training.** We give an explicit construction showing that a single transformer block, run autoregressively over a structured CoT trace, simulate $L$ steps of gradient descent for training an $N$-layer neural network (Section 4).

- **Dynamic Masking as a Selective Exposure Mechanism.** We identify redundancy as the core bottleneck in prior ICGD simulations of deep networks. Dynamic masking removes this bottleneck and yields an $O(N)$ computation improvement over Wu et al. (2025).

- **Extensions to Fundamental Statistical Models.** We establish approximation guarantees for multi-step optimization on generalized linear models, ridge regression, and lasso regression using constant depth softmax

attention/transformer modules with CoT and bounded-context access (Section 3).

**Related Work and Organization.** We defer detailed related work to Appendix B. Section 2 introduces the CoT setting and notation. Section 3 shows CoT with dynamic masking in a simplest setting; sliding window on basic statistical models. Section 4 presents our main theorem and construction for deep networks. Proofs are in Appendix C, and numerical experiments are in Appendix E.

## 2 Preliminaries and Problem Setup

In this section, we establish the formal setup of the Chain-of-Thought In-Context Gradient Descent (CoT-ICGD) framework we analyze in this paper. To provide enough background, We introduce the concepts of (i) in-context learning, (ii) in-context gradient descent, (iii) CoT-ICGD setup, and finally (iv) gradient-descent computations for deep neural networks.

**In-Context Learning.** We follow the notation from Wu et al. (2025) in this section. Consider a dataset $\mathcal{D}_n \coloneqq \{(x_i, y_i)\}_{i \in [n]}$ drawn i.i.d. from a distribution $\mathbb{P}$, where $\{x_i\}_{i \in [n]} \subseteq \mathbb{R}^d$ is the input of $f$ and $\{y_i\}_{i \in [n]} \subseteq \mathbb{R}$ is the function output. Importantly, each in-context example $(\mathcal{D}_n, x_{n+1})$ comes from a different distribution $\mathbb{P}_j$, such as a different ground truth linear model or neural network parametrized by $w_j^\star \in \mathbb{R}^d$. The in-context learning ability refers to a pretrained transformer that predict the output corresponding to a test input $x_{n+1}$ based on the in-context examples $\mathcal{D}_n$ without parameter updates across many $P_j$. Formally, we encode the context and test input $(\mathcal{D}_n, x_{n+1})$

as a prompt $Z \in \mathbb{R}^{D \times (n+1)}$

$$Z = \begin{bmatrix} x_1 & x_2 & \cdots & x_n & x_{n+1} \\ y_1 & y_2 & \cdots & y_n & 0 \\ p_1 & p_2 & \cdots & p_n & p_{n+1} \end{bmatrix} \in \mathbb{R}^{D \times (n+1)}, \quad (1)$$

where each vector $p_i \in \mathbb{R}^D$ holds any necessary auxiliary information such as positional encoding.

**In-Context Gradient Descent.** To define what we mean by a transformer implementing in-context gradient descent, we consider a parametric model $f(w, \cdot) : \mathbb{R}^{D_w} \to \mathbb{R}$, we define the empirical risk on the in-context examples $\mathcal{D}_n = \{(x_i, y_i)\}_{i=1}^n$ as

$$\mathcal{L}_n(w) = \frac{1}{2n} \sum_{i=1}^n \ell(f(w, x_i), y_i).$$

We say the transformer $\mathrm{TF}(\cdot)$ *implements in-context gradient descent* if, given the current token representation

$$z_i^{(l)} = [x_i; y_i; \bar{w}^{(l)}; p_i],$$

passing it through the transformer updates it to

$$\mathrm{TF}(z_i^{(l)}) = z_i^{(l+1)} = [x_i; y_i; \bar{w}^{(l+1)}; p_i],$$

by approximating one gradient descent step

$$\bar{w}^{(l+1)} = \bar{w} - \eta g^{(l)} + \varepsilon^{(l)},$$
$$g^{(l)} = \frac{1}{|B_l|} \sum_{i \in B_l} \nabla \ell(f(w, x_i), y_i),$$

where $\varepsilon^{(l)}$ denotes the approximation error at step $l$. Here $B_l \subseteq \{1, \ldots, n\}$ is the set of sample indices used to compute the gradient at step $l$. Taking $B_l = \{1, \ldots, n\}$ recovers gradient descent, while any $|B_l| = m < n$ yields stochastic gradient descent (in our NN setting in Section 4 we take $m = 1$).

Then we move to the core setting of this work. To implement multiple gradient-descent steps within a fixed-size transformer, we introduce the Chain-of-Thought In-Context Gradient Descent Setup.

**Definition 2.1** (Chain-of-Thought In-Context Gradient Descent Setup). *The transformer maintains a structured CoT format. At each generation step $t$, the prompt $Z_t$ equals*

$$Z_t = \begin{bmatrix} G_1 & G_2 & \cdots & G_t \end{bmatrix} \in \mathbb{R}^{d \times tn}.$$

*Here, each block $G_t \in \mathbb{R}^{d \times n}$ contains the information necessary to perform the next step in the CoT gradient-descent procedure. Depending on the scenario, this may include the current parameter update, data samples, intermediate activations (in forward propagation), gradients (in backward propagation), and other auxiliary information. At iteration $t$, given the current input $Z_t$, the transformer*

$\mathrm{TF} : \mathbb{R}^{d \times tn} \to \mathbb{R}^{d \times n}$ *generate the next block $G_{t+1}$*

$$G_{t+1} = \mathrm{TF}(Z_t).$$

*The prompt then grows autoregressively as*

$$Z_{t+1} = \begin{bmatrix} Z_t & G_{t+1} \end{bmatrix} \in \mathbb{R}^{d \times (t+1)n}.$$

**Remark 2.1** (Multiple-Token Prediction). *Our transformer generates an $n$-token block per step, following the multi-token-prediction strategy common in modern LLM architectures (Gloeckle et al., 2024; Liu et al., 2024).*

**Remark 2.2** (Explicit Form of $G_t$.). *For statistical model (Section 3), we use subscript $l$ instead of $t$ to to indicate it contains the $l$-th gradient update. Each block $G_l$ stores $[x_i; y_i; w^{(l)}; 1]_{i \in [n]}$. For deep networks, blocks alternate between forward-pass block $F_i$ and backward-pass states $B_i$ for $i \in [N]$, see Section 4 for the details.*

Finally, to prepare for subsequent theoretical analysis (Section 4), we recall the standard definitions of forward and backward computations for gradient descent in multi-layer neural networks.

**N-Layer Neural Network Gradient Descent.** We define the forward computation in Definition 2.2 and backpropagation in Definition 2.3. Later in Section 4, we replicate the forward and backward computations from (2)–(4) through a CoT mechanism.

**Definition 2.2** ($N$-Layer Feed-Forward Layer). *Let $x \in \mathbb{R}^d$ be the input, and let $W_i \in \mathbb{R}^{d \times d}$ denote the weights of the $i$-th layer for $i \in [N]$. Denote the activation function as $\sigma(\cdot)$. Define pre-activation $z_i$ and activation $o_i$ as*

$$z_i := W_i o_{i-1}, \quad o_i := \sigma(z_i),$$

*where we set input $o_0 = x$, and note that $o_N$ is the output of FFN.*

**Definition 2.3** (Backpropagation of Gradient Update for $N$-layer FFN). *Given target label $y$, denote the loss function as $\ell := \ell(o_N, y)$. The backpropagation computes the gradient in a recursive manner as follows. First, compute the error at the output layer:*

$$\delta_N = \nabla_{o_N} \ell \odot \sigma'_N(z_N). \quad (2)$$

*Second, compute $\delta_{N-1}, \cdots, \delta_1$ by propogating the error backwards. For $i = N - 1, N - 2, \ldots, 1$, propagate the error backwards:*

$$\delta_i = \underbrace{W_{i+1}^\top \delta_{i+1}}_{d \times 1} \odot \underbrace{\sigma'_i(z_i)}_{d \times 1}. \quad (3)$$

*Third, compute gradients with respect to weights:*

$$\frac{\partial \ell}{\partial W_i} = \delta_i o_{i-1}^\top \quad (4)$$

**Notations.** We use lower case letters denote (column) vectors and upper case letters denote matrices. let $n$ denote the number of in-context examples and $d$ the feature dimension. For an $N$-layer feed-forward network, we let $W_1, \ldots, W_N$, each in $\mathbb{R}^{d \times d}$, denote the layer weights. For statistical model, we denote $w^{(l)} \in \mathbb{R}^d$ after the $l$-th gradient update, where $l = 1, \ldots, L$ and $L$ is the total number of ICGD updates. We use $0_d \in \mathbb{R}^d$ to denote the zero column vector and use $1_d \in \mathbb{R}^d$ to to denote the one column vector. We also use $e_i^{(d)} \in \mathbb{R}^d$ to denote one-hot vector whose $i$-th is the only non-zero entry and is equal to 1.

**Attention.** Lastly, we define the Softmax attention layer considered in this work.

**Definition 2.4** (Attention). *Let $H$ denotes the number of heads. For input sequence $X \in \mathbb{R}^{d \times n}$, we define the multi-head attention layer as*

$$\text{Attn}_{\text{m}}(X)$$
$$= \sum_{h=1}^{H} W_V^{(h)} X \, \text{Softmax}((W_K^{(h)} X)^\top W_Q^{(h)} X) W_O^{(h)},$$

*where $W_K^{(h)}, W_Q^{(h)} \in \mathbb{R}^{d_h \times d}, W_V^{(h)} \in R^{d_o \times d}, W_O^{(h)} \in \mathbb{R}^{n \times n_o}$ for $h \in [H]$. We use $\text{Attn}_s$ to denote single-head attention.*

Following Hu et al. (2025; 2026), we pick non-identical hidden dimensions for $W_K, W_Q, W_V, W_O$ for the generality of the analysis.

## 3 CoT Enables Single-Layer ICGD of Statistical Models

In this section, we demonstrate a series of theoretical approximations power of CoT prompting in the ICGD fashion. We prove that by only *single-layer* attention (Theorem 3.1 and Theorem C.2) or *single-transformer block* (Theorem 3.2), equipped with CoT prompt, the model learn multiple fundamental statistical models through in-context gradient descent. This improve prior works (Bai et al., 2023) by $O(L)$ memory consumption, and improve the unrealistic ReLU attention to softmax attention. The statistical model includes generalize linear model, ridge regression, and lasso regression.

Each proof in this section reuses the same prompt format defined below.

**Definition 3.1** (CoT Prompt Format for Statistical Model). *At iteration $l \in [L]$, the model sees $Z_l \in \mathbb{R}^{(2d+2) \times ln}$ in the form of*

$$Z_l = \begin{bmatrix} G_1 & \cdots & G_l \end{bmatrix}$$

$$= \begin{bmatrix} x_1 & \ldots & x_n & \cdots & x_1 & \ldots & x_n \\ y_1 & \ldots & y_n & \cdots & y_1 & \ldots & y_n \\ w^{(1)} & \ldots & w^{(1)} & \ldots & w^{(l)} & \ldots & w^{(l)} \\ 1 & \ldots & 1 & \cdots & 1 & \ldots & 1 \end{bmatrix},$$

*where each block $G_l$ with $n$ columns stores one gradient-descent update.*

Considering the model already finish $l$-th update, every proof in this section follows two steps.

1. **Select** the most recent block $G_l \in \mathbb{R}^{d \times n}$ by a $n$ length context windows.

2. **Produce** next gradient update $G_l$ with one multi-head attention layer plus a token-wise linear map. we extend the (Hu et al., 2025, Theorem 4.1, In-Context Gradient Descent) to the CoT setting. According to the universal approximation power of softmax attention, we approximate gradient $\nabla_w \mathcal{L}(w)$ for any $C^1$ loss.

**Softmax Attention ICGD Approximation.** Before diving into specific statistical models, we state a foundational result that a single-layer softmax attention equipped with CoT is a universal approximator for gradient descent updates. We state the ICGD theorem in the CoT setting as follows.

**Lemma 3.1** (In-Context Gradient Descent, Theorem 4.1 of (Hu et al., 2025)). *Let $\ell : \mathbb{R} \times \mathbb{R} \to \mathbb{R}$ be any $C^1$ loss function defined on $(w^\top x_i, y_i)$. With input $X_l$ in the form of*

$$X := \begin{bmatrix} x_1 & x_2 & \cdots & x_n \\ y_1 & y_2 & \cdots & y_n \\ w^{(l)} & w^{(l)} & \cdots & w^{(l)} \\ 1 & 1 & \cdots & 1 \end{bmatrix},$$

*when $X$ is bounded, there exists a multi-head self-attention $\text{Attn}_{\text{m}}$ with skip connections and each attached with a linear layer $A$, such that for any $\epsilon > 0$, irrelevant of $X$, we have:*

$$\left\| \text{Attn}_{\text{m}} \circ A(X) - \begin{bmatrix} x_{1:n} \\ y_{1:n} \\ (w^{(l)} - \eta \nabla_w \mathcal{L}(w^{(l)})) 1_n^\top \\ 1_n^\top \end{bmatrix} \right\|_\infty \leq \epsilon,$$

*where $x_{1:n} = [x_1, \ldots, x_n], y_{1:n} = [y_1, \ldots, y_n], \eta$ denotes the learning rate and $\mathcal{L}(w) := \sum_{i=1}^{n} \ell(w^\top x_i, y_i)$ is an empirical loss upon the given input-output pairs.*

**Remark 3.1.** *Lemma 3.1 comes from (Hu et al., 2025)[Theorem 4.1]. It establishes that a single-layer softmax attention approximate one-step gradient descent update for any $C^1$ loss.*

We now demonstrate how to leverage this single-step approximation capability within our CoT framework (Definition 2.1) to perform multi-step ICGD for specific statistical

models. We begin with Generalized Linear Models, a broad and fundamental class of models whose learning often involves minimizing such $C^1$ loss functions.

### 3.1 Generalized Linear Model

Consider data $\{(x_i, y_i)\}_{i=1}^n$ with $x_i \in \mathbb{R}^d$ and $y_i \in \mathbb{R}$. Define the GLM estimator $w_{\text{GLM}}$ as

$$w_{\text{GLM}} := \underset{w \in \mathbb{R}^d}{\arg\min} \mathcal{L}_{\text{GLM}}(w)$$
$$= \underset{w \in \mathbb{R}^d}{\arg\min}(\frac{1}{n}\sum_{i=1}^n \ell_{\text{GLM}}(w^\top x_i, y_i)),$$

with $\ell_{\text{GLM}}(s, y) = -ys + \int_0^s g(t)\,dt$, where the link function $g : \mathbb{R} \to \mathbb{R}$ is continuous. Then we define a ball in $\ell_2$-norm for later use.

**Definition 3.2.** *Let $B_2^k(R)$ denote a ball containing all points in $\mathbb{R}^k$ whose $\ell_2$-norm is at most $R$.*

Then we have the following theorem.

**Standing Convention for Section 3.** Throughout Section 3, the data lie in a fixed compact set:

$$\max_{i \in [n]} \|x_i\|_\infty \le B_x, \qquad \max_{i \in [n]} |y_i| \le B_y.$$

When a CoT prompt $Z_l = [G_1, \ldots, G_l]$ enters an update map, the map reads the most recent block $G_l$ through the length-$n$ context window in Definition 3.1. We keep the notation $\text{Attn}_m \circ A(Z_l)$ for this windowed application.

**Theorem 3.1** (CoT ICGD Approximation of Generalized Linear Model). *Under the GLM setup above, assume that $\mathcal{L}_{\text{GLM}}$ is $\alpha$-strongly convex and $\beta$-smooth on $B_2^d(B_w)$, and that $\|w_{\text{GLM}}\|_2 \le B_w/2$. Initialize $w_{\text{CoT}}^{(0)} = 0_d$ and set $\eta = 1/\beta$. For any $\epsilon > 0$ and any $L \le B_w/(2\epsilon)$, there exist a fixed multi-head attention layer $\text{Attn}_m$ and a fixed linear map $A$ such that, whenever the latest block stores $w_{\text{CoT}}^{(l)}$,*

$$\left\| \text{Attn}_m \circ A(Z_l) - \begin{bmatrix} x_{1:n} \\ y_{1:n} \\ (w_{\text{CoT}}^{(l)} - \eta\nabla\mathcal{L}_{\text{GLM}}(w_{\text{CoT}}^{(l)}))1_n^\top \\ 1_n^\top \end{bmatrix} \right\|_\infty$$
$$\le \frac{\epsilon}{\sqrt{d}}.$$

*Consequently, with $\kappa = \beta/\alpha$*

$$\|w_{\text{CoT}}^{(L)} - w_{\text{GLM}}\|_2 \le L\epsilon + \exp\left(-\frac{L}{2\kappa}\right)\frac{B_w}{2}.$$

*Proof Sketch.* We first use continuity of $\partial_s\ell(s, y) = -y + g(s)$ and Lemma 3.1 to show that, a single-layer multi-head attention block with context windows of size $n$ approximates a gradient descent step within error $\epsilon$. Reusing the above attention blocks generate the next predictions

in the sense of Definition 2.1, and we use Lemma A.2 to accumulate the each step error to $L\epsilon$. Finally, by classical result the gradient descent has exponential convergence rate toward $w_{\text{GLM}}$ under strong convexity and smoothness (Lemma A.1). Combine the both error term through the triangle inequality yields the final result. See Appendix C.1 for a detailed proof. $\square$

**Remark 3.2.** *For simplicity, we drop the final in-context prediction $x_{n+1}$. This prediction is trivial by adding a 2-head attention layer constructed in (Bai et al., 2023).*

We also provide the same multi-step ICGD result on ridge regression. See Appendix C.2. It is a statistical model to prevent overfitting and a standard benchmark in studies of in-context learning (Akyürek et al., 2023; Bai et al., 2023). The proof follows the same logic as Theorem 3.1.

The previous two statistical models focus on smooth loss functions. To demonstrate a broader applicability of our single-layer CoT-ICGD setting, we next extend the result to lasso regression, which contains non-smooth $\ell_1$ regularization.

### 3.2 Lasso Regression

We consider approximating proximal gradient descent for lasso regression. Specifically, the lasso minimizer and loss is defined as

$$w_{\text{lasso}} := \underset{w \in \mathbb{R}^d}{\arg\min} \mathcal{L}_{\text{lasso}}(w)$$
$$= \underset{w \in \mathbb{R}^d}{\arg\min}(\frac{1}{2n}\sum_{i=1}^n (w^\top x_i - y_i)^2 + \lambda\|w\|_1).$$

The classical method for minimizing losses with a non-smooth regularizer (such as $\ell_1$ norm) is Proximal Gradient Descent (PGD). The single-step PGD updates parameters via standard gradient steps followed by a proximal operator $\text{prox}_{\eta\mathcal{R}}$

$$w_{\text{PGD}}^{t+1} := \text{prox}_{\eta\mathcal{R}}(w_{\text{PGD}}^t - \eta\nabla\mathcal{L}_n^0(w_{\text{PGD}}^t)),$$

where $\mathcal{L}_n^0(w) := \frac{1}{2n}\sum_{i=1}^n (w^\top x_i - y_i)^2$.

Now we state the ICGD result on lasso regression.

**Theorem 3.2** (Chain-of-Thought Proximal ICGD Approximation of Lasso Regression). *Assume that $R \ge 2\|w_{\text{lasso}}\|_2$. Let $\beta$ be a smoothness constant of $\mathcal{L}_n^0$ on $B_2^d(R)$. Initialize $w_{\text{CoT}}^{(0)} = 0_d$ and set $\eta = 1/\beta$. Let*

$$C_0 := \|\nabla\mathcal{L}_n^0(0_d)\|_2.$$

*For any $\epsilon > 0$ and any $L \le R/(2\epsilon)$, there exist a fixed multi-head attention module $\text{Attn}_m$, a fixed linear map $A$, and a fixed feed-forward network $\text{FFN}_{\text{prox}}$ such that, whenever*

the latest CoT block stores $w_{\mathrm{CoT}}^{(l)}$,

$$\left\| \mathrm{FFN}_{\mathrm{prox}} \circ \mathrm{Attn}_m \circ A(Z_l) \right.$$

$$- \left. \begin{bmatrix} x_{1:n} \\ y_{1:n} \\ \mathrm{prox}_{\eta\lambda\|\cdot\|_1}\big(w_{\mathrm{CoT}}^{(l)} - \eta\nabla\mathcal{L}_n^0(w_{\mathrm{CoT}}^{(l)})\big)1_n^\top \\ 1_n^\top \end{bmatrix} \right\|_\infty \le \frac{\epsilon}{\sqrt{d}}.$$

*Consequently,*

$$\mathcal{L}_{\mathrm{lasso}}(w_{\mathrm{CoT}}^{(L)}) - \mathcal{L}_{\mathrm{lasso}}(w_{\mathrm{lasso}})$$
$$\le (C_0 + \beta R + \lambda\sqrt{d})L\epsilon + \frac{\beta}{2}(L\epsilon)^2 + \frac{\beta R^2}{8L}.$$

*Proof Sketch.* The proof consist of two steps. First we approximate the standard gradient descent step of least square loss, and second the the proximal operator. We approximate the gradient descent step by multihead attention according to Lemma 3.1. For the proximal operator, we rely on (Bai et al., 2023, Proposition C.1) that the proximal operator for the $\ell_1$-norm regularizer is exactly approximable by FFN layer. See Appendix C.3 for a detailed proof. □

**Remark 3.3.** *Bai et al. (2023) use the ReLU attention, hence exactly construct the gradient of least-square loss $\partial_s\ell(s,t) = \partial_s\ell(w^T x_i, y_i) = \partial_s\frac{1}{2}(w^T x_i - y_i)^2 = (w^T x_i - y_i)x_i$. Replacing the ReLU attention with softmax attention inevitably introduces an approximation error $\epsilon$ at each PGD step. With sufficiently large attention width so that $\epsilon = O(1/L^2)$, we recover the standard convergence rate $O(1/L)$ of proximal gradient descent.*

# 4 CoT Gradient Descent Simulation of $N$-Layer Neural Network

In this section, we show CoT is capable of simulating more complex model in-context, the multi-step gradient descent of $N$-Layer Neural Network (Definition 2.2). We achieve this by a single-layer transformer equipped with a dynamic masking scheme. Importantly, the dynamic masking ensures the transformer attends only to the minimal, necessary information from the CoT history at each step. The proofing idea is to replicate the forward and backward computations (Definition 2.3) through attention and feed-forward layer. We provide explicit construction in Theorem 4.1 and Theorem 4.2. The final result shows that CoT reasoning handle non-convex optimisation of deep networks in context with lower resource requirements.

## 4.1 Simulating Forward and Backpropagation via CoT

We construct the $L$-step ICGD simulation of $N$-layer neural network as a CoT sequence of $2LN$ blocks. To begin with, we define some terminology for later proof.

**Definition 4.1.** *For any $W \in \mathbb{R}^{k\times d}$, define $\mathrm{vec}(\cdot)$ such that*

$$\mathrm{vec}(X) \in \mathbb{R}^{kd}, \quad (\mathrm{vec}(X))_{(j-1)\,k+i} = W_{i,j},$$

*for all $i \in [k], j \in [d]$. That is, $\mathrm{vec}(W)$ stacks the $d$ columns of $W$, each of length $k$, into a single $kd$-dimensional column vector.*

**Definition 4.2** (Rounds, Phases). *Our in-context training process of $N$-layer feed-forward network consists of $l$ repeated rounds, each round consisiting of 2 phases—the forward propagation phase and the backward propagation phase. Every $2N$ time steps consists a round, where the first $N$ steps belongs to forward propagation and last $N$ steps belong to backward propagation.*

We now define key block types used to store the state of gradient descent of the $i$-th layer of $N$-layer neural network as follow.

**Definition 4.3** (Form of Inputs and Outputs). *Let $l$ denote the current round. Denote $W_i$ as the weight of the $i$-th layer of the target $N$-layer FFN in the last round. The initial data blocks are in the form of*

$$X_l := \begin{bmatrix} x_l \cdot 1_{1\times d} \\ 0_{(3d+d^2)\times d} \\ 2 \cdot 1_{1\times d} \end{bmatrix}, \quad Y_l := \begin{bmatrix} y_l \cdot 1_{1\times d} \\ 0_{(2d+d^2)\times d} \\ 2_{1\times d} \end{bmatrix},$$

*for all $l \in [L]$[1,2]. The transformer generates the forward block $F_i$ and backward block $B_i$ defined as*

$$F_i := \underbrace{\begin{bmatrix} o_i \cdot 1_{1\times d} \\ o_{i-1} \cdot 1_{1\times d} \\ \sigma'(z_i) \cdot 1_{1\times d} \\ \mathrm{vec}(W_i) \cdot 1_{1\times d} \\ 1_{1\times d} \end{bmatrix}}_{(1+3d+d^2)\times d},$$

$$B_i := \underbrace{\begin{bmatrix} \delta_i \cdot 1_{1\times d} \\ W_i^\top \delta_i \cdot 1_{1\times d} \\ (\mathrm{vec}(W_i) - \eta\mathrm{vec}(\delta_i o_i^\top)) \cdot 1_{1\times d} \\ 0_{(d+1)\times d}\cdot \end{bmatrix}}_{(1+3d+d^2)\times d}.$$

---

[1]We note that the bottom row are tags to help identify the input. It let the update block in Proposition 4.3 distinguish the four routed input, and then apply the correct sub computation.

[2]In Section 3, we use $n$ to denote the number of in-context examples because each update is taken with respect to the empirical loss over all $n$ examples. In Section 4, by contrast, each round uses a single data pair $(x_l, y_l)$, so we index the data blocks by the round index $l$ and write $X_l, Y_l$.

*In the first round $l = 1$, the $B_i$ has the special form*

$$B_i := \underbrace{\begin{bmatrix} 0_{2d \times d} \\ \text{vec}(W_i) \cdot 1_{1 \times d} \\ 0_{(d+1) \times d} \end{bmatrix}}_{(1+3d+d^2) \times d}.$$

Next we introduce the design of the dynamic masking. According to Definition 2.2, to generate forward block $F_i$ we need to retrieve $F_{i-1}$ and $B_i$. For generating $B_i$, according to Definition 2.3 we need to retrieve $B_{i+1}$ and $F_i$. Hence, we design the dynamic masking as follows.

**Definition 4.4** (Dynamic Masking). *Let $t$ be the global generation step of the CoT sequence in Definition 2.1 after the initial prompt. The initialization steps are handled separately by Propositions 4.1 and 4.2. The first forward sweep using the initially supplied $B_i^{(1)}$ blocks is described in Appendix D. To generate the $t$-th block in the recurrent trajectory, the dynamic masking retrieves the pair of blocks*

$$\mathcal{M}(t) = (t - 1, \ t - 2((t-1) \bmod N) - 1)$$

*After retrieval, the two selected blocks are concatenated and fed to the corresponding one-step update block.*

**Remark 4.1** (Provably Resource Efficiency). *Instead of processing all $\{W_i\}_{1 \leq i \leq N}$ in every transformer block (Wang et al., 2024; Wu et al., 2025), dynamic masking only propagates 2 weight matrix to the downstream transformer layer. Specifically, in the proof of Wu et al. (2025), they flatten all the weight matrices $\{W_i\}_{1 \leq i \leq N}$ and encode onto each token. Hence, in the attention mechanism, the weight matrix multiplication is in the scale of $O(N)$. We instead only process 2 weight matrices at a time. This results in $O(N)$ computation saving.*

We now move to the construction transformer layer, implementing the forward and backpropagation.

**Theorem 4.1** (In-Context Calculation of Forward Propagation). *With the input it receives from the dynamic masking mechanism, for any $\epsilon > 0$, there exists a single-layer transformer network $\mathcal{F}_{\text{forward}}$ with positional encoding such that*

$$\left\| \mathcal{F}_{\text{forward}}\left( \begin{bmatrix} F_{i-1} \\ B_i \end{bmatrix} \right) - F_i \right\|_\infty \leq \epsilon,$$

*where $F_i$ and $B_i$ are the forward and backpropagation blocks defined in Definition 4.3.*

*Proof Sketch.* First, use a linear transformation to extract the needed information $o_{i-1}$ and $\text{vec}(W_i)$, and construct a single attention to approximate $o_i$. Then construct the second attention to approximate $\sigma'(z_i)$. See Appendix C.4 for a detailed proof. □

**Theorem 4.2** (In-Context Calculation of Backward Propagation). *With the input it receives from the dynamic masking mechanism, For any $\epsilon > 0$, there exists a single-layer transformer network $\mathcal{F}_{\text{backprop}}$ with positional encoding such that*

$$\left\| \mathcal{F}_{\text{backprop}}\left( \begin{bmatrix} B_{i+1} \\ F_i \end{bmatrix} \right) - B_i \right\|_\infty \leq \epsilon,$$

*where $F_i$ and $B_i$ are defined in Definition 4.3.*

*Proof Sketch.* First, a linear transformation to extract the needed information from the concatenated input $B_{i+1}$ of $F_i$. Then we construct a $\text{FFN}_1$ to compute

$$\underbrace{W_{i+1}^\top \delta_{i+1} \odot \sigma'(z)}_{\delta_i} \cdot 1_{1 \times d}.$$

Third, by constructing a multihead attention layer, we use part of the heads to multiply above $\delta_i$ with $o_{i-1}$, we get $\text{vec}(\delta_i o_{i-1}^\top) \cdot 1_{1 \times d}$. Then construct another linear transform to subtract $\text{vec}(W_i)$ from this output. Finally, use another branch of multihead attention to calculate $W_i^\top \delta_i$ and output it as the final output gives us the desired result. See Appendix C.5 for a detailed proof. □

We now construct a single layer transformer to initialize the first Forward Block as the base case of the recursive construction.

**Proposition 4.1** (Initialization of the First Forward Block). *Fix $l \in [L]$[3], and let $X_l$, $B_1^{(l)}$, and $F_1^{(l)}$ be the blocks defined in Definition 4.3. Then for any $\varepsilon > 0$, there exists a single Transformer block (with residual/skip connections) $\mathcal{F}_x$ such that, for the input obtained by concatenating the two blocks,*

$$Z = \begin{bmatrix} B_1^{(l)} \\ X_l \end{bmatrix},$$

*the output $\mathcal{F}_x(Z)$ approximates the desired forward block $F_1^{(l)}$ such that*

$$\left\| \mathcal{F}_x(Z) - F_1^{(l)} \right\|_\infty \leq \varepsilon.$$

*Proof.* See Appendix C.6 for detailed proof. □

After completing the full forward computation, we construct a corresponding block to initialize the first backward-propogation.

**Proposition 4.2** (Initialization the Backward Block). *Fix $l \in [L]$, and let $F_N^{(l)}$, $Y_l$, and $B_N^{(l+1)}$ be the blocks defined in Definition 4.3. Then for any $\epsilon > 0$, there exists a single Transformer block (with residual/skip connections) $\mathcal{F}_y$ such*

*that, for the input obtained by concatenating the two blocks,*

$$Z = \begin{bmatrix} F_N^{(l)} \\ Y_l \end{bmatrix},$$

*the output $\mathcal{F}_y(Z)$ approximates the desired backward block $B_N^{(l+1)}$ such that*

$$\|\mathcal{F}_y(Z) - B_N^{(l+1)}\|_\infty \le \epsilon.$$

*Proof.* See Appendix C.7 for detailed proof. □

To avoid manually switching between different block updates, we next show that a single Transformer block can automatically implement the correct one step updated for any valid input.

**Proposition 4.3** (Universal One-Step Update Block). *Fix $\varepsilon > 0$. There exists a single-layer Transformer block $\mathrm{TF} : \mathbb{R}^{D \times d} \to \mathbb{R}^{D \times d}$ with the following property.*

*Let $\mathcal{Z}$ denote the set of valid routed inputs $Z \in \mathbb{R}^{D \times d}$ that arise from the block formats in Definition 4.3, i.e., $Z$ is exactly one of the four types:* forward-type, backward-type, *$x$-type, or $y$-type. Let $M > 0$ be any constant such that $\|Z\|_\infty \le M/2$ for all $Z \in \mathcal{Z}$.*

*Then for every $Z \in \mathcal{Z}$,*

$$\|\mathrm{TF}(Z) - T(Z)\|_\infty \le \varepsilon,$$

*where $T$ is the corresponding target one-step update map:*

$$T(Z) = \begin{cases} T_{\mathrm{f}}(Z), & Z \text{ is forward-type,} \\ T_{\mathrm{b}}(Z), & Z \text{ is backward-type,} \\ T_x(Z), & Z \text{ is } x\text{-type,} \\ T_y(Z), & Z \text{ is } y\text{-type.} \end{cases}$$

*Proof.* See Appendix C.8 for detailed proof. □

Until now we equip ourself all the building blocks to simulate gradient descent of $N$-layer neural network. Now we formally state the input format and, how to combine those blocks into single transformer and generate the ICGD trajectory of neural network.

**Full Input Format.** Represent the prompt as a token matrix $Z \in \mathbb{R}^{D \times d(2L+N)}$. The initial context is the block concatenation

$$Z^{(1)} = \begin{bmatrix} X_1 & Y_1 & \cdots & X_L & Y_L & B_1^{(1)} & \cdots & B_N^{(1)} \end{bmatrix},$$

where $X_\ell, Y_\ell$ ,and $B_1^{(1)}, \ldots, B_N^{(1)} \in \mathbb{R}^{D \times d}$ are defined ion Definition 4.3. We start the CoT generation after this input format. We use $l$ to index the generated block. See

Appendix D for an example of a generated CoT ICGD trajectory.

The following proposition states the approximation error of gradient update's final trajectory.

**Proposition 4.4** (In-context GD State Trajectory via Single Transformer Block). *Fix integers $N \ge 1$ and $L \ge 1$, and a labeled sequence $(x_l, y_l)_{l=1}^L$ with input formatting as above. Let $\varepsilon_{\mathrm{C1}}, \varepsilon_{4.1}, \varepsilon_{\mathrm{C2}}, \varepsilon_{4.2} > 0$ be the accuracy parameters from Proposition 4.1, Theorem 4.1, Proposition 4.2, and Theorem 4.2, and let $\varepsilon_{\mathrm{C3}} > 0$ be the routing accuracy parameter from Proposition 4.3. Define*

$$\varepsilon_{\mathrm{round}} := \varepsilon_{\mathrm{C1}} + (N-1)\varepsilon_{4.1} + \varepsilon_{\mathrm{C2}} + (N-1)\varepsilon_{4.2} + (2N+1)\varepsilon_{\mathrm{C3}}.$$

*Assume that the exact round update map is $K$-Lipschitz in $\|\cdot\|_{\infty,\max}$. Then there exists a single-layer Transformer block $\mathrm{TF}$ defined in Proposition 4.3, such that it produces state blocks $\{\widehat{B}_i^{(l)}\}_{i \in [N], l=1}^{L+1}$ satisfying $\widehat{B}_i^{(1)} = B_i^{(1)}$ and, for every $l \in [L]$,*

$$\max_{i \in [N]} \|\widehat{B}_i^{(l+1)} - B_i^{(l+1)}\|_\infty \le \varepsilon_{\mathrm{round}} \sum_{t=0}^{l-1} K^t.$$

*In particular,*

$$\max_{i \in [N]} \|\widehat{B}_i^{(L+1)} - B_i^{(L+1)}\|_\infty \le \varepsilon_{\mathrm{round}} \sum_{t=0}^{L-1} K^t.$$

*Proof Sketch.* See Appendix C.9 for a detailed proof. □

**Remark 4.2.** *The dynamic masking mechanism (Definition 4.4) ensures that at each of these $2LN$ steps, the fixed-depth transformer only processes a small and constant number of relevant prior blocks. This methodology prevent the need to carry all information through every layer of a deep transformer and the input-scaling issue in prior ICL works when simulating the training of complex model in-context.*

## 5 Discussion and Conclusion

We demonstrate a one-layer transformer, with CoT and dynamic masking, simulates multi-step gradient descent in-context in a more efficient way compared to prior work (Section 4). With the CoT nature of recursive generation, we demonstrate its memory saving through the application of ICGD on the statistical model (Section 3). Combined with dynamic masking, the model only needs to process the necessary information from the CoT sequence, and leads to $O(N)$ computational savings compared to prior results on simulating the training of a deep network in-context. In summary, we provide theoretical guarantee on the effectiveness and efficiency of CoT with dynamic masking. This paves the way for enabling transformers to perform complex, multi-step learning algorithms in context with better resource utilization.

**Connecting to Practice.** The idea of generating very long CoT but tackling the computation issue (quadratic cost of attention) has raised the notice of the community (Giannou et al., 2023; Yang et al., 2025). To produce an infinite-length CoT process to mimic an infinite thinking process, we must find some way to keep only the necessary information to produce the next token. The dynamic masking implements this spirit. Recently, Yang et al. (2025) approach the same intuition as us to solve this problem, but with different method and aspect of theoretical analysis. Specifically, they introduce a reduction rule that deletes token spans once they become irrelevant, and empirically show that a limited-size and context window transformer solves challenging tasks, such as Einstein's puzzle, given a long thinking time. This result backs up the usefulness of our dynamic masking intuition. In contrast to (Yang et al., 2025), our dynamic masking keeps the token in the input sequence, but only lets the attention layer see the necessary tokens. Hence our method doesn't have the risk of deleting information later might be useful, but still eliminates the growth of computation cost along with longer CoT. The theoretical axis we investigate is also different. They prove that a fixed-size and context window transformer generates a token trace that encodes the Turing machine. Complementary to the Turing machine, we show the strength of this architecture from the *in-context learning* perspective with an optimization objective in deep learning and machine learning models. Our theoretical analysis tackles this by proving CoT simulating training deep networks and statistical models, and gives an explicit construction of such transformer and provable approximation error.

**Limitations and Future Work.** Our guarantees rest on several idealized conditions. First, dynamic masking is treated as cost-free and omniscient. Ee ignore the overhead of computing masks and the latency of repeatedly scanning an ever-growing prompt. Second, the claimed $O(N)$ FLOP saving is asymptotic—the constant factors introduced by serial reuse of the single block and by prompt growth remain unquantified. Lastly, our proofs target ReLU feed-forward networks and convex statistical objectives. Extending the framework to convolutional, residual, or attention-based settings and to adaptive or second-order optimizers is left for future work.

## Acknowledgments

The authors thank Mingcheng Lu, Mimi Gallagher, Sara Sanchez, Dino Feng and Andrew Chen for enlightening discussions; Maojiang Su, Yi-Chen Lee, Po-Chiao Lin, and Jennifer Zhang for collaborations on related topics and pointing out typos. JH also thanks the Red Maple Family for support. The authors would like to thank the anonymous reviewers and program chairs for constructive comments.

JH is partially supported by the Walter P. Murphy Fellowship and the Terminal Year Fellowship (Paul K. Richter Memorial Award) of Northwestern University. Han Liu is partially supported by NIH R01LM1372201, NSF AST-2421845, Simons Foundation MPS-AI-00010513, AbbVie, Dolby and Chan Zuckerberg Biohub Chicago Spoke Award. This research was supported in part through the computational resources and staff contributions provided for the Quest high performance computing facility at Northwestern University which is jointly supported by the Office of the Provost, the Office for Research, and Northwestern University Information Technology. The content is solely the responsibility of the authors and does not necessarily represent the official views of the funding agencies.

## Impact Statement

By the theoretical nature of this work, we do not anticipate any negative social impact.

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

# Appendix

## A   Supplementary Theoretical Backgrounds

Here we introduce some axillary lemmas and definitions for later convenience.

**Lemma A.1** (Proposition A.2 of (Bai et al., 2023), Gradient Descent for Smooth and Strongly Convex Functions)**.** *Let $L$ be $\alpha$-strongly convex and $\beta$-smooth function. Then the gradient descent $w_{\mathrm{GD}}^{t+1} = w_{\mathrm{GD}}^t - \eta \nabla L(w_{\mathrm{GD}}^t)$ with learning rate $\eta = 1/\beta$ satisfies the following two exponential convergence rates*

$$\|w_{\mathrm{GD}}^t - w^\star\|_2^2 \leq \exp\left(-\frac{t}{\kappa}\right) \cdot \|w_{\mathrm{GD}}^0 - w^\star\|_2^2,$$

$$L(w_{\mathrm{GD}}^t) - L(w^\star) \leq \frac{\beta}{2} \exp\left(-\frac{t}{\kappa}\right) \cdot \|w_{\mathrm{GD}}^0 - w^\star\|_2^2,$$

*where $\kappa := \beta/\alpha$ is the condition number and $w^\star$ is the minimizer of $L$.*

**Definition A.1** ($B_2^k(R)$)**.** *Let $B_2^k(R)$ denote a ball containing all points in $\mathbb{R}^k$ whose $\ell_2$-norm is at most $R$.*

**Lemma A.2** (Lemma 14 of (Bai et al., 2023), Composition of Error for Approximating Convex Gradient Descent)**.** *Suppose $f : \mathbb{R}^d arrow \mathbb{R}$ is a convex function. Define optimal solution $w^\star := \operatorname{argmin}_{w \in \mathbb{R}^d} f(w)$, $R \geq 2\|w^\star\|_2$, and assume that $f$ is $\beta$-smooth on $\mathbb{B}_2^d(R)$. We define sequences $\{\widehat{w}^l\}_{l \geq 0} \in \mathbb{R}^d$, with $\{w_{\mathrm{GD}}^l\}_{l \geq 0} \in \mathbb{R}^d$ as follow:*

$$\begin{cases} \widehat{w}^{l+1} = \widehat{w}^l - \eta \nabla f(\widehat{w}^l) + \epsilon^l, & \|\epsilon^l\|_2 \leq \varepsilon, \\ w_{\mathrm{GD}}^{l+1} = w_{\mathrm{GD}}^l - \eta \nabla f(w_{\mathrm{GD}}^l), \end{cases} \quad \text{for all} \quad l \geq 0,$$

*where $\epsilon^l$ is the l-th iteration approximation error and is bounded by $\varepsilon$. Then as long as learning rate $\eta \leq 2/\beta$, for any iteration $L \leq R/(2\varepsilon)$ we have:*

$$\|\widehat{w}^L - w_{\mathrm{GD}}^L\|_2 \leq L\varepsilon,$$

*this means that the approximation error is linear with the number of iterations. We also have:*

$$\|\widehat{w}^L\|_2 \leq \frac{R}{2} + L\varepsilon \leq R.$$

*This ensures the norm of approximation solution $\widehat{w}^L$ remain bounded by $R$.*

**Lemma A.3** (Proposition A.3 of (Bai et al., 2023), Proximal gradient Descent for Convex Function)**.** *Let $L = f + h :$*

$\mathbb{R}^d \to \mathbb{R}$, where $f$ is convex and $\beta$-smooth for some $\beta \geq 0$, and $h$ is a convex function. Then the proximal gradient descent

$$w_{\text{PGD}}^{t+1} = \text{prox}_{\eta\mathcal{R}}(w_{\text{PGD}}^t - \eta\nabla L(w_{\text{PGD}}^t))$$

with $\eta = 1/\beta$ satisfies the following three for $t \geq 1$:

- *The sequence $\{L(w_{\text{PGD}}^t)\}$ is decreasing.*

- *For any minimizer $w^\star \in \arg\min_{w\in\mathbb{R}^d} L(w)$,*

$$L(w_{\text{GD}}^{t+1}) - L(w^\star) \leq \frac{\beta}{2}(\|w_{\text{PGD}}^t - w^\star\|_2^2 - \|w_{\text{PGD}}^{t+1} - w^\star\|_2^2),$$

  *and since $L(w_{\text{PGD}}^{t+1}) \geq L(w^\star)$ we have*

$$\|w_{\text{PGD}}^{t+1} - w^\star\|_2 \leq \|w_{\text{PGD}}^t - w^\star\|_2.$$

- *For $k \geq 1$, $t \geq 0$, we have*

$$L(w_{\text{PGD}}^{t+k}) - L(w^\star) \leq \frac{\beta}{2k}\|w_{\text{PGD}}^t - w^\star\|_2^2.$$

**Lemma A.4** (Lemma C.1 of (Bai et al., 2023), Composition of error for approximating convex PGD). *Suppose $f$ : $\mathbb{R}^d arrow \mathbb{R}$ is a convex function and $\mathcal{R}$ is a convex regularizer. Define optimal solution $w^\star := \text{argmin}_{w\in\mathbb{R}^d} f(w) + \mathcal{R}(w)$, $R \geq 2\|w^\star\|_2$, and assume that $f$ is $\beta$-smooth on $\mathbb{B}_2^d(R)$. We define the sequences transformer approximated weights update as $\{\widehat{w}^l\}_{l\geq 0} \in \mathbb{R}^d$ and the standard gradient descent as $\{w_{\text{PGD}}^l\}_{l\geq 0} \in \mathbb{R}^d$:*

$$\begin{cases} \widehat{w}^{l+1} = \text{prox}_{\eta\mathcal{R}}(\widehat{w}^l - \eta\nabla f(\widehat{w}^l)) + \epsilon^l, & \|\epsilon^l\|_2 \leq \varepsilon, \\ w_{\text{GD}}^{l+1} = \text{prox}_{\eta\mathcal{R}}(w_{\text{GD}}^l) - \eta\nabla f(w_{\text{GD}}^l), \end{cases} \quad \text{for all} \quad l \geq 0,$$

*where $\epsilon^l$ is the $l$-th iteration approximation error and is bounded by $\varepsilon$. Then as long as learning rate $\eta \leq 2/\beta$, for any iteration $L \leq R/(2\varepsilon)$ we have*

$$\|\widehat{w}^L - w_{\text{GD}}^L\|_2 \leq L\varepsilon.$$

*This means the approximation error is linear with the number of iterations. We also have*

$$\|\widehat{w}^L\|_2 \leq \frac{R}{2} + L\varepsilon \leq R.$$

*This ensures the norm of approximation solution $\widehat{w}^L$ remain bounded by $R$.*

**Lemma A.5** (Theorem 3.1 of Hu et al. (2025), Attention Approximates Truncated Linear Models In-Context). *Fix real numbers $a < b$. Let the input be*

$$X = \begin{bmatrix} x_1 & x_2 & \cdots & x_n \\ w & w & \cdots & w \\ t_1 & t_2 & \cdots & t_n \end{bmatrix} \in \mathbb{R}^{(2d+1)\times n},$$

*where $\{w, x_i\}_{i\in[n]}$ are bounded. For a precision parameter $p > n$, there exists a single-layer, single-head self-attention* Attn *with a linear transformation $A : \mathbb{R}^{(2d+1)\times n} \to \mathbb{R}^{(2d+d_o+2)\times p}$, such that* Attn $\circ A : \mathbb{R}^{d\times n} \to \mathbb{R}^{d_o\times n}$ *satisfies, for any $i \in [n]$,*

$$\|\text{Attn} \circ A(X)_{:,i} - \text{Range}_{[a,b]}(w_i^\top x_i + t_i)e_{\widetilde{k_i}}\|_\infty$$

$$\leq \underbrace{\max\{|a|, |b|\} \cdot \epsilon_0}_{\text{finite-}\beta \text{ softmax error}} + \underbrace{\frac{b-a}{p}}_{\text{interpolation error}}.$$

*Here $e_{\widetilde{k}_i}$ is a one-hot vector with a value of $1$ at the $\widetilde{k}_i$-th index and $0$ elsewhere, $\epsilon_0 = O(e^{-\beta\delta})$, and $\widetilde{k}_i \in [d_o]$ is*

$$k_i = \operatorname*{argmin}_{k \in \{0,1,\cdots,p-1\}} (-2x_i^\top w_i - 2t_i + \widetilde{L}_0 + \widetilde{L}_k) \cdot k,$$

$$\widetilde{k}_i = G(k_i).$$

*Here $G : [p] \to [d_o]$ denotes any set-to-set function sending each selected interpolation index $k_i$ into an appropriate integer $\widetilde{k}_i \in [d_o]$ for $i \in [n]$.*

**Remark A.1.** *Lemma A.5 prove attention mechanism approximation ReLU activation in-context. Hence by multi-head attention, we add up each head's output to approximate the computation of $N$-layer ReLU neural network.*

# B Related Work

**Theoretical Analysis of Chain-of-Thought.** Feng et al. (2023) use circuit complexity theory to prove that when solving basic arithmetic problems, a depth-limited transformer need super-polynomially growing model size with respect to input length, while an autoregressive Transformer of constant size solves the tasks by generating CoT in a formal math language. Li et al. (2023) prove transformer in-context learns a $L$-layer FFN's output with arbitrary error when incorporating the intermediate output of multiple layer FFN into the prompt, which they call CoT prompt. Specifically, they formulate multilayer FFN as a compositional function $f := f_L \circ \cdots f_1$. With input $x_i$, denote intermediate output of each layer as $s_i^l = f_l(s_i^{l-1})$, the CoT prompt with $n$ sample is in the form $(x_1, s_1^1, \cdots, s_1^L; \cdots, x_n, s_n^1, \cdots ; s_n^L)$. By leveraging a filtering mechanism to isolate the tokens relevant to each layer, the transformer decouples the task into simpler linear regression subproblems for each $f_l$. Kim & Suzuki (2025) prove when training a one-layer transformer with ground-truth intermediate steps, it solve the hard $k$-parity problem in single gradient update. Huang et al. (2025) show that the global minimizer of one-layer linear transformer trained on CoT-style loss for linear regression performs multi-step gradient descent. Unlike this work, we focus on a problem different from the training dynamic when equipped transformer with CoT objectives: a theoretical proof of a *softmax* transformer with CoT input simulates multi-step gradient descent on various statistical model losses, and how dynamic-masking further improves the efficiency of inference time CoT. Prystawski et al. (2023) study CoT from a probabilistic perspective. They formulate the reasoning process of CoT as a sequential process of estimating conditional probabilities of intermediate variables. When the training data reveals local structure, they show CoT performs better than direct inference. Cui et al. (2024) study CoT by comparing it to In-context learning. Their results show that under assumptions on model parameters, CoT achieves lower error compared to ICL. Further, they also show that CoT is more sensitive to noise being introduced to intermediate steps compared to ICL. Li et al. (2024); Merrill & Sabharwal (2024) both study CoT from the perspective of circuit complexity. Merrill & Sabharwal (2023) show that the expressive power of constant-depth transformers is bounded, and CoT provides higher expressiveness by breaking the limitation of constant depth.

**In-Context Gradient Descent of Neural Network.** Despite (Wu et al., 2025), several other works investigate the transformer's ability to simulate gradient descent training of neural networks through in-context. Early studies focused on simple models. For example, Akyürek et al. (2023) demonstrate a transformer is capable of performing gradient updates for linear regression in an implicit fashion. Similarly, Dai et al. (2023) argue GPT-style models function as meta-optimizers performing gradient descent. Subsequent works further extend these insights. Von Oswald et al. (2023) provide a constructive proof that a transformer is capable of carrying out multiple steps of gradient descent internally. Bai et al. (2023) further show that transformers can even select among different learning algorithms in context (hence earning the moniker "transformers as statisticians"), provably adapting to the best solver for a given task. Importantly, Bai et al. (2023) also show that a frozen transformer imitates gradient descent for two-layer networks, but their technique does not extend to deeper models because it avoids explicit backpropagation. In contrast, we prove this in a CoT manner that the one-layer transformer simulates $L$ GD steps on $N$-layer network. This is the first formal evidence that the "thinking steps" written in a CoT prompt implement back-propagation. By establishing this result, we clarify how LLMs might learn complex learning algorithms by CoT prompting. Giannou et al. (2023) treat transformer as a looped computer and proves by recursively using 13 transformer layers, it simulates training of $N$-layer sigmoid neural network. In contrast, our paper uses only one-layer transformer block to approximate training of $N$-layer ReLU neural network, which is also more practical in real-world use of neural network activation (Glorot et al., 2011; Maas et al., 2013; Ramachandran et al., 2018). Also, the input sequence length for the looped transformer in Giannou et al. (2023) increase with the number of network layers $N$. This is because executing more sequential operations (more forward and backward steps for deeper networks during gradient descent) requires an increased number of "instruction" in the input sequence by their design. Gatmiry et al. (2024) further study the training dynamic of linear looped transformer, showing that the global minimizer on the empirical risk implements multi-step preconditioned gradient descent on the linear regression instance. In contrast, we study constructive expressivity of a practical softmax Transformer. We show that a fixed Transformer, run autoregressively with CoT traces, implements multi-step ICGD for multiple statistical objectives (GLM, ridge, lasso) and simulates full forward–backpropagation for $N$-layer neural network.

**CoT with Dynamic Masking: Efficiency over (Wu et al., 2025).** Wu et al. (2025) show transformers can simulate the training of an $N$-layer neural network in-context. Their method needs a gigantic prompt, encoding all weights and activations at every layer. This causes redundant computation because all information moves through each layer. Input length also grows with the simulated model size. This limits the complexity of models learned in-context with fixed input size. We address these issues by using chain-of-thought (CoT) prompting with dynamic masking. Instead of stacking many transformer layers, we reuse one transformer block across $L$ gradient descent steps. At each step, we mask the transformer's

attention. Each attention head only sees tokens needed for the current update. This prevents unnecessary information from moving forward. Our method reduces computation by $O(N)$ compared to (Wu et al., 2025). It simulates training an $N$-layer network with constant transformer depth (i.e., $O(1)$ depth in a loop and reused repeatedly), not depth $O(N)$. This also removes the fixed-input-size limitation. The prompt can grow with CoT, enabling more complex models and longer training.

# C Proofs of Main Text

## C.1 Proof of Theorem 3.1

Consider data $\{(x_i, y_i)\}_{i=1}^n$ with $x_i \in \mathbb{R}^d$ and $y_i \in \mathbb{R}$. Define the empirical loss

$$\mathcal{L}_{\mathrm{GLM}}(w) = \frac{1}{n} \sum_{i=1}^n \ell_{\mathrm{GLM}}(w^\top x_i, y_i), \quad \text{with} \quad \ell_{\mathrm{GLM}}(s, y) = -ys + \int_0^s g(t)dt,$$

where the link function $g : \mathbb{R} \to \mathbb{R}$ is continuous.

Define the input as:

$$X := \begin{bmatrix} x_1 & x_2 & \cdots & x_n \\ w_{\mathrm{CoT}}^{(0)} & w_{\mathrm{CoT}}^{(0)} & \cdots & w_{\mathrm{CoT}}^{(0)} \\ y_1 & y_2 & \cdots & y_n \\ 1 & 1 & \cdots & 1 \end{bmatrix},$$

where $w_{\mathrm{CoT}}^{(0)} = 0_d \in \mathbb{R}^d$ is the initialization of the generalized linear model. We set $Z_1 = X$.

**Theorem C.1** (Restated of Theorem 3.1, Chain-of-Thought Gradient Descent Approximation of GLM.). *Under the GLM setup above, assume that $\mathcal{L}_{\mathrm{GLM}}$ is $\alpha$-strongly convex and $\beta$-smooth on $B_2^d(B_w)$, and that $\|w_{\mathrm{GLM}}\|_2 \leq B_w/2$. Initialize $w_{\mathrm{CoT}}^{(0)} = 0_d$ and set $\eta = 1/\beta$. For any $\epsilon > 0$ and any $L \leq B_w/(2\epsilon)$, there exist a fixed multi-head attention layer $\mathrm{Attn}_m$ and a fixed linear map $A$ such that, whenever the latest block stores $w_{\mathrm{CoT}}^{(l)}$,*

$$\left\| \mathrm{Attn}_m \circ A(Z_l) - \begin{bmatrix} x_{1:n} \\ y_{1:n} \\ (w_{\mathrm{CoT}}^{(l)} - \eta \nabla \mathcal{L}_{\mathrm{GLM}}(w_{\mathrm{CoT}}^{(l)}))1_n^\top \\ 1_n^\top \end{bmatrix} \right\|_\infty$$

$$\leq \frac{\epsilon}{\sqrt{d}}.$$

*Consequently, with $\kappa = \beta/\alpha$*

$$\|w_{\mathrm{CoT}}^{(L)} - w_{\mathrm{GLM}}\|_2 \leq L\epsilon + \exp\left(-\frac{L}{2\kappa}\right)\frac{B_w}{2}.$$

*Proof.* Our proof consists eight conceptual steps.

**Step 1: Choose the Attention Accuracy.** Let

$$\epsilon_{\mathrm{att}} := \frac{\epsilon}{\sqrt{d}}.$$

This choice turns a weight-block $\ell_\infty$ error into an $\ell_2$ weight error of size at most $\epsilon$.

**Step 2: Match the Loss Format in Lemma 3.1.** Define

$$\widetilde{\ell}_{\mathrm{GLM}}(s, y) := \frac{1}{n} \ell_{\mathrm{GLM}}(s, y).$$

Then

$$\sum_{i=1}^n \widetilde{\ell}_{\mathrm{GLM}}(w^\top x_i, y_i) = \mathcal{L}_{\mathrm{GLM}}(w).$$

Since $g$ is continuous, $\widetilde{\ell}_{\mathrm{GLM}}$ is $C^1$ in its first argument. Hence Lemma 3.1 applies to $L_{\mathrm{GLM}}$ on the compact domain fixed in Appendix E.

**Step 3: Approximate One GD Step.** By Lemma 3.1, there exist a fixed multi-head attention layer $\mathrm{Attn}_m$ and a fixed linear map $A$ such that, for every $w \in B_2^d(B_w)$,

$$\left\| \mathrm{Attn}_m \circ A \begin{bmatrix} x_{1:n} \\ y_{1:n} \\ w 1_n^\top \\ 1_n^\top \end{bmatrix} - \begin{bmatrix} x_{1:n} \\ y_{1:n} \\ (w - \eta \nabla \mathcal{L}_{\mathrm{GLM}}(w)) 1_n^\top \\ 1_n^\top \end{bmatrix} \right\|_\infty$$

$$\leq \epsilon_{\mathrm{att}}. \hspace{4cm} \text{(By Lemma 3.1)}$$

The module is fixed after we choose $(B_x, B_y, B_w, \epsilon_{\mathrm{att}})$. It does not depend on the CoT step $l$.

**Step 4: Insert the Module into the CoT Window.** At step $l$, the latest block $G_l$ stores $w_{\mathrm{CoT}}^{(l)}$. By the Appendix E standing convention, the update map reads $G_l$ from $Z_l$ through the length-$n$ context window. Hence

$$\left\| \mathrm{Attn}_m \circ A(Z_l) - \begin{bmatrix} x_{1:n} \\ y_{1:n} \\ (w_{\mathrm{CoT}}^{(l)} - \eta \nabla \mathcal{L}_{\mathrm{GLM}}(w_{\mathrm{CoT}}^{(l)})) 1_n^\top \\ 1_n^\top \end{bmatrix} \right\|_\infty$$

$$\leq \epsilon_{\mathrm{att}}. \hspace{4cm} \text{(By Step 3)}$$

**Step 5: Convert Block Error to Weight Error.** Extract $w_{\mathrm{CoT}}^{(l+1)}$ from any column of the generated weight block. Then there exists $e_l \in \mathbb{R}^d$ such that

$$w_{\mathrm{CoT}}^{(l+1)} = w_{\mathrm{CoT}}^{(l)} - \eta \nabla \mathcal{L}_{\mathrm{GLM}}(w_{\mathrm{CoT}}^{(l)}) + e_l.$$

Moreover,

$$\begin{aligned} \|e_l\|_2 &\leq \sqrt{d} \|e_l\|_\infty \\ &\leq \sqrt{d} \epsilon_{\mathrm{att}} \\ &= \epsilon. \hspace{3cm} \text{(By the choice of } \epsilon_{\mathrm{att}}) \end{aligned}$$

**Step 6: Compare CoT with Exact GD.** Define exact gradient descent by

$$\begin{aligned} w_{\mathrm{GD}}^{(0)} &= 0_d, \\ w_{\mathrm{GD}}^{(l+1)} &= w_{\mathrm{GD}}^{(l)} - \eta \nabla L_{\mathrm{GLM}}(w_{\mathrm{GD}}^{(l)}). \end{aligned}$$

Since $\mathcal{L}_{\mathrm{GLM}}$ is $\alpha$-strongly convex, it is convex. Also, $\eta = 1/\beta \leq 2/\beta$ and $L \leq B_w/(2\epsilon)$. Thus Lemma A.2 gives

$$\|w_{\mathrm{CoT}}^{(L)} - w_{\mathrm{GD}}^{(L)}\|_2 \leq L\epsilon, \hspace{3cm} \text{(By Lemma A.2)}$$

and keeps the CoT iterates in $B_2^d(B_w)$. Therefore Step 3 applies at every CoT step.

**Step 7: Use Exact GD Convergence.** Since $\eta = 1/\beta$, Lemma A.1 gives

$$\begin{aligned} \|w_{\mathrm{GD}}^{(L)} - w_{\mathrm{GLM}}\|_2 &\leq \exp\left(-\frac{L}{2\kappa}\right) \|w_{\mathrm{GLM}}\|_2 \\ &\leq \exp\left(-\frac{L}{2\kappa}\right) \frac{B_w}{2}. \hspace{2cm} \text{(Since } \|w_{\mathrm{GLM}}\|_2 \leq B_w/2) \end{aligned}$$

**Step 8: Combine the Two Errors.** Finally,

$$\begin{aligned} \|w_{\mathrm{CoT}}^{(L)} - w_{\mathrm{GLM}}\|_2 &\leq \|w_{\mathrm{CoT}}^{(L)} - w_{\mathrm{GD}}^{(L)}\|_2 + \|w_{\mathrm{GD}}^{(L)} - w_{\mathrm{GLM}}\|_2 \hspace{1cm} \text{(By triangle inequality)} \\ &\leq L\epsilon + \exp\left(-\frac{L}{2\kappa}\right) \frac{B_w}{2}. \hspace{2cm} \text{(By Steps 6 and 7)} \end{aligned}$$

This completes the proof. □

## C.2 Chain-of-Thought ICGD Approximation of Ridge Regression

We state the ridge regression as follows. Consider the ridge regression estimators $w_{\text{ridge}}^\lambda$ over the in-context examples in $\mathcal{D}$ with regularization $\lambda \geq 0$ :

$$w_{\text{ridge}}^\lambda := \underset{w \in \mathbb{R}^d}{\operatorname{argmin}} \mathcal{L}_{\text{ridge}}(W) = \underset{w \in \mathbb{R}^d}{\operatorname{argmin}} \{\frac{1}{2n} \sum_{i=1}^n (\langle w, x_i \rangle - y_i)^2 + \frac{\lambda}{2} \|w\|_2^2\}.$$

We state the CoT ICGD on ridge regression empirical risk by single attention layer.

**Theorem C.2** (Chain-of-Thought ICGD Approximation of Ridge Regression.). *Fix $\lambda \geq 0$ and assume $\|w_{\text{ridge}}^\lambda\|_2 \leq B_w/2$. Let $\eta \leq 2/\beta$ and choose the CoT length $L \leq \frac{B_w}{2}/(2\epsilon)$. Then there exists a multi-head attention layer with a linear map such that the CoT iterates satisfy*

$$\|w_{\text{CoT}}^{(L)} - w_{\text{ridge}}^\lambda\|_2 \leq L\epsilon + e^{-L/(2\kappa)} \frac{B_w}{2}.$$

*Proof.* The Hessian of ridge regression is

$$H = \frac{1}{N} X X^\top + \lambda I_d.$$

The ridge regression loss is known to be $\alpha$-strongly convex and and $\beta$-smooth with

$$\alpha = \lambda_{\min}(H), \quad \beta = \lambda_{\max}(H), \quad ,\kappa = \beta/\alpha.$$

Since $\partial \ell(s, y)/\partial_s = s - y$ is continuous for least square loss, and $\ell_2$ penalty contribute $\lambda w$ to the gradient is also $C^1$. Hence by Lemma 3.1, for every $\epsilon \geq 0$ there exists a single-layer multi-head attention with a linear projection denoted as $\text{Attn}_{\text{m}} \cdot A$ such that

$$\|\text{Attn}_m \circ A(G_l) - \begin{bmatrix} x_1 & \dots & x_n \\ y_1 & \dots & y_n \\ w_{\text{CoT}}^{(l)} - \eta \nabla_w \mathcal{L}_{\text{ridge}}(w_{\text{CoT}}^{(l)}) & \dots & w_{\text{CoT}}^{(l)} - \eta \nabla_w \mathcal{L}_{\text{ridge}}(w_{\text{CoT}}^{(l)}) \\ l+1 & \dots & l+1 \\ 1 & \dots & 1 \end{bmatrix} \|_\infty \leq \epsilon, \tag{5}$$

with learning-rate $\eta \leq 2/\beta$.

Next we bound the error composition along CoT by Lemma A.2. We write the single-step CoT gradient descent approximation as

$$w_{\text{CoT}}^{(l+1)} = w_{\text{CoT}}^{(l)} - \eta \nabla_w \mathcal{L}_{\text{ridge}}(w_{\text{CoT}}^{(l)}) + \varepsilon^{(l)},$$

where $\|\varepsilon^{(l)}\|_\infty \leq \epsilon$ according to (5). We convert the $\ell_\infty$ error into $\ell_2$ and absorb the $\sqrt{2d+3}$ into $\epsilon$ to have $\|\varepsilon^{(l)}\|_2 \leq \epsilon$.

By the Lemma A.2 and choose

$$L \leq B_w/(2\epsilon), \quad B_w \geq 2\|w_{\text{ridge}}^\lambda\|_2,$$

we have

$$\|w_{\text{CoT}}^{(L)} - w_{\text{GD}}^{(L)}\|_2 \leq L\epsilon, \quad \|w_{\text{CoT}}^{(L)}\|_2 \leq B_w. \tag{6}$$

Because $\mathcal{L}_{\text{ridge}}$ is $\alpha$-strongly convex and $\beta$-smooth, Lemma A.1 gives exponential convergence of standard gradient descent

$$\|w_{\text{GD}}^{(L)} - w_{\text{ridge}}^\lambda\|_2 \leq e^{-L/(2\kappa)} \|w_{\text{ridge}}^\lambda\|_2. \tag{7}$$

Combining (6) and (7) we have

$$\|w_{\text{CoT}}^{(L)} - w_{\text{ridge}}^{\lambda}\|_2 \le L\epsilon + e^{-L/(2\kappa)}\|w_{\text{ridge}}^{\lambda}\|_2.$$

This completes the proof. □

## C.3  Proof of Theorem 3.2

We consider approximating proximal gradient descent for lasso regression. Specifically, the lasso loss is defined as follows

$$\mathcal{L}_{\text{lasso}} = \frac{1}{2n} \sum_{i=1}^{n} (w^{\top} x_i - y_i)^2 + \lambda \|W\|_1.$$

The classical method for minimizing losses with a non-smooth regularizer (such as $\ell_1$ norm) is Proximal Gradient Descent (PGD). The single-step PGD updates parameters via stardard gradient steps followed by a proximal operator $\text{prox}_{\eta \mathcal{R}}$

$$w_{\text{PGD}}^{t+1} := \text{prox}_{\eta \mathcal{R}}(w_{\text{PGD}}^t - \eta \nabla \mathcal{L}_n^0(w_{\text{PGD}}^t)), \quad \text{where} \quad \mathcal{L}_n^0(w) := 1/2n \sum_{i=1}^{n} (w^{\top} x_i - y_i)^2. \tag{8}$$

**Theorem C.3** (Restated of Theorem 3.2, Chain-of-Thought Proximal ICGD Approximation of Lasso Regression). *Assume that $R \geq 2\|w_{\text{lasso}}\|_2$. Let $\beta$ be a smoothness constant of $\mathcal{L}_n^0$ on $B_2^d(R)$. Initialize $w_{\text{CoT}}^{(0)} = 0_d$ and set $\eta = 1/\beta$. Let*

$$C_0 := \|\nabla \mathcal{L}_n^0(0_d)\|_2.$$

*For any $\epsilon > 0$ and any $L \leq R/(2\epsilon)$, there exist a fixed multi-head attention module $\text{Attn}_m$, a fixed linear map $A$, and a fixed feed-forward network $\text{FFN}_{\text{prox}}$ such that, whenever the latest CoT block stores $w_{\text{CoT}}^{(l)}$,*

$$\|\text{FFN}_{\text{prox}} \circ \text{Attn}_m \circ A(Z_l)$$

$$- \begin{bmatrix} x_{1:n} \\ y_{1:n} \\ \text{prox}_{\eta \lambda \|\cdot\|_1} \left( w_{\text{CoT}}^{(l)} - \eta \nabla \mathcal{L}_n^0(w_{\text{CoT}}^{(l)}) \right) 1_n^{\top} \\ 1_n^{\top} \end{bmatrix} \|_{\infty} \leq \frac{\epsilon}{\sqrt{d}}.$$

*Consequently,*

$$\mathcal{L}_{\text{lasso}}(w_{\text{CoT}}^{(L)}) - \mathcal{L}_{\text{lasso}}(w_{\text{lasso}})$$

$$\leq (C_0 + \beta R + \lambda \sqrt{d})L\epsilon + \frac{\beta}{2}(L\epsilon)^2 + \frac{\beta R^2}{8L}.$$

*Proof.* Let

$$f(w) := \mathcal{L}_n^0(w),$$
$$h(w) := \lambda \|w\|_1,$$
$$S_{\eta \lambda}(u) := \text{prox}_{\eta h}(u) = \text{prox}_{\eta \lambda \|\cdot\|_1}(u).$$

**Step 1: Choose the Error Budget.**  Let

$$\epsilon_{\text{att}} := \frac{\epsilon}{\sqrt{d}}.$$

This choice turns a weight-block $\ell_{\infty}$ error into an $\ell_2$ weight error of size at most $\epsilon$.

**Step 2: Match the Smooth Loss to Lemma 3.1.**  Define

$$\widetilde{\ell}(s, y) := \frac{1}{2n}(s - y)^2.$$

Then

$$\sum_{i=1}^{n} \widetilde{\ell}(w^{\top} x_i, y_i) = \frac{1}{2n} \sum_{i=1}^{n} (w^{\top} x_i - y_i)^2$$

$$= \mathcal{L}_n^0(w).$$

Also,

$$\partial_s \widetilde{\ell}(s, y) = \frac{1}{n}(s - y).$$

Thus $\widetilde{\ell}$ is $C^1$ in its first argument. Hence Lemma 3.1 applies to $\mathcal{L}_n^0$ on the compact domain fixed in Appendix E.

**Step 3: Approximate the Gradient Step.** For any $w \in B_2^d(R)$, define

$$G(w) := \begin{bmatrix} x_{1:n} \\ y_{1:n} \\ w 1_n^\top \\ 1_n^\top \end{bmatrix}.$$

By Lemma 3.1, there exist a fixed multi-head attention module $\text{Attn}_m$ and a fixed linear map $A$ such that, for every $w \in B_2^d(R)$,

$$\left\| \text{Attn}_m \circ A(G(w)) - \begin{bmatrix} x_{1:n} \\ y_{1:n} \\ (w - \eta \nabla \mathcal{L}_n^0(w)) 1_n^\top \\ 1_n^\top \end{bmatrix} \right\|_\infty$$

$$\le \epsilon_{\text{att}}. \hspace{5cm} \text{(By Lemma 3.1)}$$

After we choose $(R, \lambda, \eta, \epsilon_{\text{att}})$, this module is fixed. It does not depend on $l$ or on the current weight $w$.

**Step 4: Apply the Proximal Map.** By Bai et al. (2023, Proposition C.1), a ReLU FFN exactly implements the soft-thresholding map $S_{\eta\lambda}$. We embed this FFN into the weight rows and copy the other rows. The copy operation uses the identity

$$z = \text{ReLU}(z) - \text{ReLU}(-z).$$

Thus there exists a fixed token-wise FFN $\text{FFN}_{\text{prox}}$ that maps the weight rows through $S_{\eta\lambda}$ and leaves the other rows unchanged.

The soft-thresholding map is 1-Lipschitz in $\ell_\infty$:

$$\|S_{\eta\lambda}(u) - S_{\eta\lambda}(v)\|_\infty \le \|u - v\|_\infty.$$

Therefore Step 3 gives, for every $w \in B_2^d(R)$,

$$\left\| \text{FFN}_{\text{prox}} \circ \text{Attn}_m \circ A(G(w)) - \begin{bmatrix} x_{1:n} \\ y_{1:n} \\ S_{\eta\lambda}(w - \eta \nabla \mathcal{L}_n^0(w)) 1_n^\top \\ 1_n^\top \end{bmatrix} \right\|_\infty$$

$$\le \epsilon_{\text{att}}. \hspace{3cm} \text{(By Step 3 and nonexpansiveness of } S_{\eta\lambda})$$

**Step 5: Insert the Module into the CoT Window.** At step $l$, the latest block stores $w_{\text{CoT}}^{(l)}$. By the Appendix E standing convention, the update map reads the latest length-$n$ block from $Z_l$. Hence

$$\left\| \text{FFN}_{\text{prox}} \circ \text{Attn}_m \circ A(Z_l) - \begin{bmatrix} x_{1:n} \\ y_{1:n} \\ S_{\eta\lambda}(w_{\text{CoT}}^{(l)} - \eta \nabla \mathcal{L}_n^0(w_{\text{CoT}}^{(l)})) 1_n^\top \\ 1_n^\top \end{bmatrix} \right\|_\infty$$

$$\le \epsilon_{\text{att}}. \hspace{5cm} \text{(By Step 4)}$$

**Step 6: Convert Block Error to Weight Error.** Extract $w_{\text{CoT}}^{(l+1)}$ from any column of the generated weight block. Then there exists $e_l \in \mathbb{R}^d$ such that

$$w_{\text{CoT}}^{(l+1)} = S_{\eta\lambda}\big(w_{\text{CoT}}^{(l)} - \eta\nabla\mathcal{L}_n^0(w_{\text{CoT}}^{(l)})\big) + e_l.$$

Moreover,

$$
\begin{aligned}
\|e_l\|_2 &\leq \sqrt{d}\|e_l\|_\infty \\
&\leq \sqrt{d}\epsilon_{\text{att}} \\
&= \epsilon. && \text{(By the choice of } \epsilon_{\text{att}})
\end{aligned}
$$

**Step 7: Compare CoT and Exact PGD.** Define the exact PGD sequence by

$$
\begin{aligned}
w_{\text{PGD}}^{(0)} &= 0_d, \\
w_{\text{PGD}}^{(l+1)} &= S_{\eta\lambda}\big(w_{\text{PGD}}^{(l)} - \eta\nabla\mathcal{L}_n^0(w_{\text{PGD}}^{(l)})\big).
\end{aligned}
$$

Since $f = \mathcal{L}_n^0$ is convex and $\beta$-smooth on $B_2^d(R)$, $\eta = 1/\beta \leq 2/\beta$, and $L \leq R/(2\epsilon)$, Lemma A.4 gives

$$\|w_{\text{CoT}}^{(L)} - w_{\text{PGD}}^{(L)}\|_2 \leq L\epsilon. \qquad \text{(By Lemma A.4)}$$

Lemma A.4 also keeps the CoT iterates in $B_2^d(R)$. Therefore Steps 3–5 apply at every CoT step.

**Step 8: Bound the Exact PGD Loss.** Since $\eta = 1/\beta$, Lemma A.3 gives

$$
\begin{aligned}
\mathcal{L}_{\text{lasso}}(w_{\text{PGD}}^{(L)}) - \mathcal{L}_{\text{lasso}}(w_{\text{lasso}}) &\leq \frac{\beta}{2L}\|w_{\text{PGD}}^{(0)} - w_{\text{lasso}}\|_2^2 \\
&= \frac{\beta}{2L}\|w_{\text{lasso}}\|_2^2 \\
&\leq \frac{\beta R^2}{8L}. && \text{(Since } R \geq 2\|w_{\text{lasso}}\|_2)
\end{aligned}
$$

Also, Lemma A.3 gives

$$\|w_{\text{PGD}}^{(L)} - w_{\text{lasso}}\|_2 \leq \|w_{\text{PGD}}^{(0)} - w_{\text{lasso}}\|_2.$$

Thus

$$
\begin{aligned}
\|w_{\text{PGD}}^{(L)}\|_2 &\leq \|w_{\text{PGD}}^{(L)} - w_{\text{lasso}}\|_2 + \|w_{\text{lasso}}\|_2 \\
&\leq 2\|w_{\text{lasso}}\|_2 \\
&\leq R.
\end{aligned}
$$

**Step 9: Transfer the Loss Bound to CoT.** Let

$$\Delta := w_{\text{CoT}}^{(L)} - w_{\text{PGD}}^{(L)}.$$

By Step 7,

$$\|\Delta\|_2 \leq L\epsilon.$$

Since $\mathcal{L}_n^0$ is $\beta$-smooth on $B_2^d(R)$,

$$
\begin{aligned}
\mathcal{L}_n^0(w_{\text{CoT}}^{(L)}) - \mathcal{L}_n^0(w_{\text{PGD}}^{(L)}) &\leq \nabla\mathcal{L}_n^0(w_{\text{PGD}}^{(L)})^\top\Delta + \frac{\beta}{2}\|\Delta\|_2^2 \\
&\leq \|\nabla\mathcal{L}_n^0(w_{\text{PGD}}^{(L)})\|_2\|\Delta\|_2 + \frac{\beta}{2}\|\Delta\|_2^2. && \text{(By Cauchy–Schwarz)}
\end{aligned}
$$

The gradient term satisfies

$$\|\nabla \mathcal{L}_n^0(w_{\mathrm{PGD}}^{(L)})\|_2 \leq \|\nabla \mathcal{L}_n^0(0_d)\|_2 + \|\nabla \mathcal{L}_n^0(w_{\mathrm{PGD}}^{(L)}) - \nabla \mathcal{L}_n^0(0_d)\|_2$$
$$\leq C_0 + \beta \|w_{\mathrm{PGD}}^{(L)}\|_2$$
$$\leq C_0 + \beta R. \qquad \text{(By Step 8)}$$

Therefore

$$\mathcal{L}_n^0(w_{\mathrm{CoT}}^{(L)}) - \mathcal{L}_n^0(w_{\mathrm{PGD}}^{(L)}) \leq (C_0 + \beta R)L\epsilon + \frac{\beta}{2}(L\epsilon)^2.$$

For the regularizer,

$$\lambda \|w_{\mathrm{CoT}}^{(L)}\|_1 - \lambda \|w_{\mathrm{PGD}}^{(L)}\|_1 \leq \lambda \|\Delta\|_1$$
$$\leq \lambda \sqrt{d} \|\Delta\|_2$$
$$\leq \lambda \sqrt{d} L\epsilon.$$

Combining the smooth loss and regularizer gives

$$\mathcal{L}_{\mathrm{lasso}}(w_{\mathrm{CoT}}^{(L)}) - \mathcal{L}_{\mathrm{lasso}}(w_{\mathrm{PGD}}^{(L)}) \leq (C_0 + \beta R + \lambda \sqrt{d})L\epsilon + \frac{\beta}{2}(L\epsilon)^2.$$

**Step 10: Combine the Bounds.**    Finally,

$$\mathcal{L}_{\mathrm{lasso}}(w_{\mathrm{CoT}}^{(L)}) - \mathcal{L}_{\mathrm{lasso}}(w_{\mathrm{lasso}})$$
$$= \left(\mathcal{L}_{\mathrm{lasso}}(w_{\mathrm{CoT}}^{(L)}) - \mathcal{L}_{\mathrm{lasso}}(w_{\mathrm{PGD}}^{(L)})\right) + \left(\mathcal{L}_{\mathrm{lasso}}(w_{\mathrm{PGD}}^{(L)}) - \mathcal{L}_{\mathrm{lasso}}(w_{\mathrm{lasso}})\right)$$
$$\leq (C_0 + \beta R + \lambda \sqrt{d})L\epsilon + \frac{\beta}{2}(L\epsilon)^2 + \frac{\beta R^2}{8L}. \qquad \text{(By Steps 8 and 9)}$$

This completes the proof. $\qquad \square$

## C.4   Proof of Theorem 4.1

**Theorem C.4** (In-context Calculation of Forward Propagation). *Let $l$ denote the current round. When $t \bmod 2N \in [N]$, the network performs forward propagation for the $N$-layer feed-forward network. The input it receives from the dynamic masking mechanism is*

$$\begin{bmatrix} F_{i-1} \\ B_i \end{bmatrix} = \begin{bmatrix} o_{i-1} \cdot 1_{1\times d} \\ o_{i-2} \cdot 1_{1\times d} \\ \sigma'(z_{i-1}) \cdot 1_{1\times d} \\ \mathrm{vec}(W_i) \cdot 1_{1\times d} \\ 1_{1\times d} \\ \delta_i \cdot 1_{1\times d} \\ (W_i^{l-1})^\top \delta_i \cdot 1_{1\times d} \\ \underbrace{(\mathrm{vec}(W_i^{l-1}) - \eta\,\mathrm{vec}(\delta_i o_i^\top))}_{\mathrm{vec}(W_i^l)} \cdot 1_{1\times d} \\ \mathrm{vec}(W_{i-1}) \cdot 1_{1\times d} \\ 0_{(d+1)\times d} \end{bmatrix}.$$

*For any $\epsilon > 0$, there exists a single-layer transformer network $\mathcal{F}_{\mathrm{forward}}$ with positional encoding such that*

$$\left\| \mathcal{F}_{\mathrm{forward}}\left(\begin{bmatrix} F_{i-1} \\ B_i \end{bmatrix}\right) - F_i \right\|_\infty \leq \epsilon,$$

*where*

$$F_i = \begin{bmatrix} o_i \cdot 1_{1\times d} \\ o_{i-1} \cdot 1_{1\times d} \\ \sigma'(z_i) \cdot 1_{1\times d} \\ \mathrm{vec}(W_i) \cdot 1_{1\times d} \\ 1_{1\times d} \end{bmatrix}.$$

*The guarantee holds except for an arbitrarily small region in the input space.*

*Proof.* Our goal is to explicitly construct a *single layer transformer block* that maps the input $\begin{bmatrix} F_{i-1} \\ B_i \end{bmatrix}$ to (an approximation of) $F_i$. Essentially, the construction is organized to mirror the forward update we aim to simulate. We will do this in the following manner:

1. extract from the input the two quantities needed to form the pre-activation $z_i$;

2. use $d$ attention heads to compute (approximately) the broadcasted pre-activation $z_i \cdot \mathbf{1}_{1\times d}$;

3. use a token-wise FFN[4] to produce the two nonlinear rows $\sigma(z_i) \cdot \mathbf{1}_{1\times d}$ and $\sigma'(z_i) \cdot \mathbf{1}_{1\times d}$ from this approximation;

4. route the remaining (linear) rows to the output via skip connections, and then bound the total $\ell_\infty$ error.

**Step 1: Extract the needed rows by a token wise linear map.**   We first apply a token wise linear transformation that extracts the rows containing $o_{i-1}$ and $\mathrm{vec}(W_i)$.

$$A\left(\begin{bmatrix} F_{i-1} \\ B_i \end{bmatrix}\right) = \begin{bmatrix} o_{i-1} \cdot \mathbf{1}_{1\times d} \\ \mathrm{vec}(W_i) \cdot \mathbf{1}_{1\times d} \end{bmatrix} \in \mathbb{R}^{(d+d^2)\times d}. \tag{9}$$

One explicit choice is the block matrix

$$A = \begin{bmatrix} I_d & 0 & 0 & 0 \\ 0 & 0 & I_{d^2} & 0 \end{bmatrix},$$

where $0$ denotes zero blocks of the appropriate dimensions.

---

[4]We note that the ReLU activation assumption is not essential.

**Step 2: Use attention to compute $z_i \cdot \mathbf{1}_{1 \times d}$.** Our next goal is to produce the pre activation $z_i \cdot \mathbf{1}_{1 \times d}$, since then the transformer FFN can apply $\sigma(\cdot)$ entrywise to obtain $o_i \cdot \mathbf{1}_{1 \times d} = \sigma(z_i) \cdot \mathbf{1}_{1 \times d}$.

Fix $h \in [d]$. To apply Lemma A.5 we need a length-$d$ vector derived from $W_i$. Let $S_h \in \mathbb{R}^{d \times d^2}$ be a fixed selection matrix such that

$$S_h \text{vec}(W_i) = w_{i,h} \in \mathbb{R}^d, \tag{10}$$

where $w_{i,h}$ satisfies

$$(z_i)_h = w_{i,h}^\top o_{i-1}. \tag{11}$$

(Equivalently, $w_{i,h}$ is the appropriate row/column slice of $W_i$ consistent with the definition of $z_i$.) Since $S_h$ is linear, we may absorb it into the head's linear $Q/K/V$ projections by composition; thus the resulting module is still a valid attention head.

By Lemma A.5, for any $\epsilon_1 > 0$ there exists an attention head $\text{Attn}_h^{(1)}$ (with positional encodings) such that

$$\|\text{Attn}_h^{(1)}(\begin{bmatrix} o_{i-1} \cdot \mathbf{1}_{1 \times d} \\ \text{vec}(W_i) \cdot \mathbf{1}_{1 \times d} \end{bmatrix}) - (z_i)_h e_h^{(d)} \cdot \mathbf{1}_{1 \times d}\|_\infty \leq \epsilon_1.$$

That is, head $h$ approximates the $h$-th coordinate block $(z_i)_h e_h^{(d)}$ ( across all tokens). Summing these $d$ heads reconstructs the full vector $z_i$ (across all tokens):

$$\|\sum_{h=1}^d \text{Attn}_h^{(1)}(\begin{bmatrix} o_{i-1} \cdot \mathbf{1}_{1 \times d} \\ \text{vec}(W_i) \cdot \mathbf{1}_{1 \times d} \end{bmatrix}) - z_i \cdot \mathbf{1}_{1 \times d}\|_\infty$$

$$= \|\sum_{h=1}^d \text{Attn}_h^{(1)}(\begin{bmatrix} o_{i-1} \cdot \mathbf{1}_{1 \times d} \\ \text{vec}(W_i) \cdot \mathbf{1}_{1 \times d} \end{bmatrix}) - \sum_{h=1}^d (z_i)_h e_h^{(d)} \cdot \mathbf{1}_{1 \times d}\|_\infty$$

$$\leq \sum_{h=1}^d \|\text{Attn}_h^{(1)}(\begin{bmatrix} o_{i-1} \cdot \mathbf{1}_{1 \times d} \\ \text{vec}(W_i) \cdot \mathbf{1}_{1 \times d} \end{bmatrix}) - (z_i)_h e_h^{(d)} \cdot \mathbf{1}_{1 \times d}\|_\infty \leq d\epsilon_1. \qquad \text{(By triangle inequality)}$$

For convenience, we name the aggregated attention output (for this specific input) as

$$Z_{\text{att}} := \sum_{h=1}^d \text{Attn}_h^{(1)} \circ A(\begin{bmatrix} F_{i-1} \\ B_i \end{bmatrix}), \tag{12}$$

where $A$ is the linear transformation from Step 1, $Z_{\text{att}} \approx z_i \cdot \mathbf{1}_{1 \times d}$ and $\|Z_{\text{att}} - z_i \cdot \mathbf{1}_{1 \times d}\|_\infty \leq d\epsilon_1$.

**Step 3: Use a token-wise FFN to produce $\sigma(z_i)$ and $\sigma'(z_i)$.** Given $Z_{\text{att}} \approx z_i \cdot \mathbf{1}_{1 \times d}$, we next produce

$$o_i \cdot \mathbf{1}_{1 \times d} = \sigma(z_i) \cdot \mathbf{1}_{1 \times d} \quad \text{and} \quad \sigma'(z_i) \cdot \mathbf{1}_{1 \times d},$$

by applying suitable token-wise feed-forward maps to $Z_{\text{att}}$.

We first construct a token-wise feed-forward network $\text{FFN}_1$ dedicated to approximating $\sigma'(\cdot)$:

$$\text{FFN}_1(Z) := \frac{1}{\delta_1}(\sigma(Z) - \sigma(Z - \delta_1 \mathbf{1}_{d \times d})), \tag{13}$$

where $\delta_1 > 0$ is chosen according to the required precision, $Z$ is a generic input to the FFN, and $\sigma(\cdot)$ is applied entrywise. Equivalently, for each $(r, c)$, (here $r$ indexes feature coordinates and $c$ indexes token positions.)

$$(\text{FFN}_1(Z))_{r,c} = \begin{cases} 1, & Z_{r,c} > \delta_1, \\ Z_{r,c}/\delta_1, & 0 < Z_{r,c} \leq \delta_1, \\ 0, & Z_{r,c} \leq 0. \end{cases}$$

In particular, whenever $Z_{r,c} \notin (0, \delta_1]$ we have $(\mathrm{FFN}_1(Z))_{r,c} = \sigma'(Z_{r,c})$. Thus, since the attention module produces $Z_{\mathrm{att}} \approx z_i \cdot \mathbf{1}_{1 \times d}$, we obtain

$$\mathrm{FFN}_1(Z_{\mathrm{att}}) \approx \sigma'(z_i) \cdot \mathbf{1}_{1 \times d},$$

except on the usual small region where some coordinate lies in $(0, \delta_1]$.

Next, we put the two nonlinear rows inside the same FFN sublayer output via a block stacking FFN:

$$\mathrm{FFN}_2(Z) := \begin{bmatrix} \sigma(Z) \\ 0_{d \times d} \\ \mathrm{FFN}_1(Z) \\ 0_{d^2 \times d} \\ \mathbf{1}_{1 \times d} \end{bmatrix}, \tag{14}$$

where the last row $\mathbf{1}_{1 \times d}$ is produced by biases (with zero input weights). $\mathrm{FFN}_2$ is still an FFN because FFNs are closed under parallelization and post composition with affine maps.

**Step 4: Route the remaining rows via a skip connection.**    The remaining rows $o_{i-1} \cdot \mathbf{1}_{1 \times d}$ and $\mathrm{vec}(W_i) \cdot \mathbf{1}_{1 \times d}$ are provided by a skip connection from $A(\cdot)$ using a token wise linear map $A_1$ that places these blocks into the correct output rows:

$$A_1(Z) := \begin{bmatrix} 0_{d \times d} & 0_{d \times d^2} \\ I_d & 0_{d \times d^2} \\ 0_{d \times d} & 0_{d \times d^2} \\ 0_{d^2 \times d} & I_{d^2} \\ 0_{1 \times d} & 0_{1 \times d^2} \end{bmatrix} Z.$$

**Step 5: Assemble the transformer block and bound the error.**    Combining the skip connected linear part and the FFN output from the attention result, the overall single-layer transformer block can be written as

$$\mathcal{F}_{\mathrm{forward}}(Z) = A_1 \circ A(Z) + \mathrm{FFN}_2(\sum_{h=1}^{d} \mathrm{Attn}_h^{(1)} \circ A(Z)). \tag{15}$$

Therefore, on the input $\begin{bmatrix} F_{i-1} \\ B_i \end{bmatrix}$ the output equals (approximately)

$$\begin{bmatrix} o_i \cdot \mathbf{1}_{1 \times d} \\ o_{i-1} \cdot \mathbf{1}_{1 \times d} \\ \sigma'(z_i) \cdot \mathbf{1}_{1 \times d} \\ \mathrm{vec}(W_i) \cdot \mathbf{1}_{1 \times d} \\ \mathbf{1}_{1 \times d} \end{bmatrix}, \qquad \text{where } o_i = \sigma(z_i),$$

except on an arbitrarily small region induced by the approximation in (13).

We now bound the $l_\infty$ error. Since the other rows are matched *exactly* by $A_1 \circ A(\cdot)$ and biases, the $l_\infty$ error reduces to the maximum of the only two nontrivial rows:

$$\|\mathcal{F}_{\mathrm{forward}}(\begin{bmatrix} F_{i-1} \\ B_i \end{bmatrix}) - F_i\|_\infty = \max\{\|\sigma(Z_{\mathrm{att}}) - o_i \cdot \mathbf{1}_{1 \times d}\|_\infty, \|\mathrm{FFN}_1(Z_{\mathrm{att}}) - \sigma'(z_i) \cdot \mathbf{1}_{1 \times d}\|_\infty\}$$

$$= \max\{\|\sigma(Z_{\mathrm{att}}) - \sigma(z_i \cdot \mathbf{1}_{1 \times d})\|_\infty, \|\mathrm{FFN}_1(Z_{\mathrm{att}}) - \sigma'(z_i) \cdot \mathbf{1}_{1 \times d}\|_\infty\} \quad \text{(By Definition 2.2)}$$

$$\leq \max\{\|Z_{\mathrm{att}} - z_i \cdot \mathbf{1}_{1 \times d}\|_\infty, \|\mathrm{FFN}_1(Z_{\mathrm{att}}) - \sigma'(z_i) \cdot \mathbf{1}_{1 \times d}\|_\infty\},$$

where the inequality uses the elementary fact that for ReLU, $|\sigma(a) - \sigma(b)| \leq |a - b|$ holds entrywise.

By the construction of the $d$ attention heads (Lemma 4.1 applied per head with accuracy $\epsilon_1$), we have

$$\Delta := \|Z_{\mathrm{att}} - z_i \cdot \mathbf{1}_{1 \times d}\|_\infty \leq d\epsilon_1. \tag{16}$$

**Controlling the** $\text{FFN}_1$ **term via a bad region.** Define the bad region

$$\mathcal{B} := \{ \begin{bmatrix} F_{i-1} \\ B_i \end{bmatrix} : \exists r \in [d] \text{ s.t. } (z_i)_r \in (-\Delta, \ \delta_1 + \Delta] \}. \tag{17}$$

We claim that for every input outside $\mathcal{B}$,

$$\text{FFN}_1(Z_{\text{att}}) = \sigma'(z_i) \cdot \mathbf{1}_{1 \times d}, \qquad \text{and hence} \qquad \|\text{FFN}_1(Z_{\text{att}}) - \sigma'(z_i) \cdot \mathbf{1}_{1 \times d}\|_\infty = 0. \tag{18}$$

To prove the claim, fix an input outside $\mathcal{B}$ and any entry $(r, c) \in [d] \times [d]$. Since the input is outside $\mathcal{B}$, either $(z_i)_r \leq -\Delta$ or $(z_i)_r \geq \delta_1 + \Delta$. In the first case, using (16) we have $(Z_{\text{att}})_{r,c} \leq (z_i)_r + \Delta \leq 0$, and by the definition of $\text{FFN}_1$ in (13), $(\text{FFN}_1(Z_{\text{att}}))_{r,c} = 0 = \sigma'((z_i)_r)$. In the second case, again by (16) we have $(Z_{\text{att}})_{r,c} \geq (z_i)_r - \Delta \geq \delta_1$, and by (13), $(\text{FFN}_1(Z_{\text{att}}))_{r,c} = 1 = \sigma'((z_i)_r)$.

Since this holds for every $(r, c)$, we conclude (18).

Therefore, for inputs outside $\mathcal{B}$,

$$\|\mathcal{F}_{\text{forward}}(\begin{bmatrix} F_{i-1} \\ B_i \end{bmatrix}) - F_i\|_\infty \leq \max\{\|\sigma(Z_{\text{att}}) - \sigma(z_i \cdot \mathbf{1}_{1 \times d})\|_\infty, 0\}$$

$$\leq \|Z_{\text{att}} - z_i \cdot \mathbf{1}_{1 \times d}\|_\infty = \Delta \leq d\epsilon_1.$$

Choosing $\epsilon_1 = \epsilon/d$ yields the desired bound $\leq \epsilon$ outside the bad region $\mathcal{B}$. Moreover, recalling $\Delta := \|Z_{\text{att}} - z_i \cdot \mathbf{1}_{1 \times d}\|_\infty \leq d\epsilon_1$, the definition (17) shows that $\mathcal{B}$ consists exactly of inputs for which some coordinate $(z_i)_r$ lies in the margin interval $(-\Delta, \ \delta_1 + \Delta]$, whose width is $\delta_1 + 2\Delta$. Hence, as $\delta_1 \to 0$ and $\epsilon_1 \to 0$ (so $\Delta \to 0$), this excluded region shrinks to the threshold sets $\{(z_i)_r = 0\}$ and $\{(z_i)_r = \delta_1\}$ in the $z_i$ coordinates. $\qquad \square$

**Remark C.1** (Activation in the Constructed Transformer FFN). *Our construction uses ReLU FFNs to implement the local nonlinear maps needed for the simulated ReLU network, including the activation, its derivative, and the gating operations in backpropagation and routing. This is a proof convenience rather than an architectural restriction. Since the proof only evaluates these submodules on bounded domain, standard universal approximation results imply that FFNs with non-polynomial activations, such as GeLU or SwiGLU, approximate these maps to error $\epsilon_{\text{act}}$ (Leshno et al., 1993). This replacement adds at most $C\epsilon_{\text{act}}$ to the one-step approximation error, which is absorbed into the final error bound in Proposition 4.4.*

## C.5 Proof of Theorem 4.2

**Theorem C.5** (Restated of Theorem 4.2, In-context Calculation of Backward Propagation). *Let $l$ denote the current round. When $t \bmod 2N \in \{N+1, \cdots, 2N\}$, the network performs backward propagation for the $N$-layer feed-forward network. The input it receives from the dynamic masking mechanism is*

$$\begin{bmatrix} B_{i+1} \\ F_i \end{bmatrix} = \begin{bmatrix} \delta_{i+1} \cdot 1_{1 \times d} \\ W_{i+1}^\top \delta_{i+1} \cdot 1_{1 \times d} \\ (\mathrm{vec}(W_{i+1}) - \eta \mathrm{vec}(\delta_{i+1} o_{i+1}^\top)) \cdot 1_{1 \times d} \\ 0_{(d+1) \times d} \\ o_i \cdot 1_{1 \times d} \\ o_{i-1} \cdot 1_{1 \times d} \\ \sigma'(z_i) \cdot 1_{1 \times d} \\ \mathrm{vec}(W_i) \cdot 1_{1 \times d} \\ 1_{1 \times d} \end{bmatrix},$$

*For any $\epsilon > 0$, there exists a single-layer transformer network $\mathcal{F}_{\mathrm{backprop}}$ with positional encoding such that*

$$\|\mathcal{F}_{\mathrm{backprop}}(\begin{bmatrix} B_{i+1} \\ F_i \end{bmatrix}) - B_i\|_\infty \leq \epsilon,$$

*where The guarantee holds except for an arbitrarily small region in the input space.*

$$B_i = \begin{bmatrix} \delta_i \cdot 1_{1 \times d} \\ W_i^\top \delta_i \cdot 1_{1 \times d} \\ (\mathrm{vec}(W_i) - \eta \mathrm{vec}(\delta_i o_i^\top)) \cdot 1_{1 \times d} \\ 0_{(d+1) \times d} \end{bmatrix}.$$

*The guarantee holds except for an arbitrarily small region in the input space.*

*Proof.* **Roadmap.** Starting from the masked input $\begin{bmatrix} B_{i+1} \\ F_i \end{bmatrix}$, we proceed in six steps:

1. Extract the specific slots needed for the $i$-th backprop update.

2. Compute the backprop signal $\delta_i = (W_{i+1}^\top \delta_{i+1}) \odot \sigma'(z_i)$ via a token-wise FFN.

3. Construct the vectorized gradient $\mathrm{vec}(\delta_i o_{i-1}^\top)$ via $d$ attention heads.

4. Update $\mathrm{vec}(W_i)$ by summing a skip branch and a scaled attention branch (implementing gradient descent).

5. Compute $W_i^\top \delta_i$ via another attention branch.

6. Assemble all outputs into $B_i$ and bound the total error.

**Step 1: Isolating the terms needed for the $i$-th backward step:** We first apply a token-wise linear map $A^*$ that extracts (and broadcasts across columns) the entries we will use in the subsequent computation:

$$A^*(\begin{bmatrix} B_{i+1} \\ F_i \end{bmatrix}) = \begin{bmatrix} \delta_{i+1} \cdot 1_{1 \times d} \\ W_{i+1}^\top \delta_{i+1} \cdot 1_{1 \times d} \\ o_{i-1} \cdot 1_{1 \times d} \\ \sigma'(z_i) \cdot 1_{1 \times d} \\ \mathrm{vec}(W_i) \cdot 1_{1 \times d} \end{bmatrix}.$$

One explicit construction is the following matrix:

$$A^*(Z) := \begin{bmatrix} I_d & 0_{d \times d} & 0_{d \times (d^2 + 2d + 1)} & 0_{d \times (2d + d^2)} \\ 0_{d \times d} & I_d & 0_{d \times (d^2 + 2d + 1)} & 0_{d \times (2d + d^2)} \\ 0_{(2d + d^2) \times d} & 0_{(2d + d^2) \times d} & 0_{(2d + d^2) \times (d^2 + 2d + 1)} & I_{2d + d^2} \end{bmatrix} Z.$$

**Step 2: Computing the ReLU backprop term $\delta_i$:** We now construct a token-wise feed-forward network that computes

$$\delta_i = l(W_{i+1}^\top \delta_{i+1} r) \odot \sigma'(z_i) \in \mathbb{R}^d$$

and broadcasts it as $\delta_i \cdot 1_{1 \times d}$. Outside an arbitrarily small region of the input space where some coordinate of $z_i$ lies in the transition interval of $\sigma'$, we have $\sigma'(z_i) \in \{0, 1\}^d$. Fix any constant

$$B_1 \geq l \| W_{i+1}^\top \delta_{i+1} r \|_\infty.$$

For each coordinate index $r \in [d]$, we aim to output

$$(\delta_i)_r = l(W_{i+1}^\top \delta_{i+1} r)_r \cdot \sigma'(z_i)_r.$$

Define the token wise FFN $\mathrm{FFN}_3 : \mathbb{R}^{(4d+d^2) \times d} \to \mathbb{R}^{d \times d}$ by

$$\mathrm{FFN}_3(Z) := \sigma(\begin{bmatrix} 0_{d \times d} & I_d & 0_{d \times d} & 2B_1 I_d & 0_{d \times d^2} \end{bmatrix} Z - B_1 \cdot 1_{d \times d}) - B_1 \cdot \begin{bmatrix} 0_{d \times d} & 0_{d \times d} & 0_{d \times d} & I_d & 0_{d \times d^2} \end{bmatrix} Z,$$

where $\sigma(\cdot)$ is ReLU applied entrywise.

Since each extracted vector is broadcast across columns, the output of $\mathrm{FFN}_3 \circ A^*(\cdot)$ has identical columns. Substituting the output of $A^*$ gives

$$\mathrm{FFN}_3 \circ A^*(\begin{bmatrix} B_{i+1} \\ F_i \end{bmatrix}) = (\sigma(W_{i+1}^\top \delta_{i+1} + 2B_1 \sigma'(z_i) - B_1 \cdot 1_d) - B_1 \sigma'(z_i)) \cdot 1_{1 \times d}.$$

Fix $r \in [d]$. Since $B_1 \geq \| W_{i+1}^\top \delta_{i+1} \|_\infty$, we have $l(W_{i+1}^\top \delta_{i+1} r)_r \in [-B_1, B_1]$. For $\sigma'(z_i)_r \in \{0, 1\}$,

$$[\sigma(W_{i+1}^\top \delta_{i+1} + 2B_1 \sigma'(z_i) - B_1 \cdot 1_d) - B_1 \sigma'(z_i)]_r = \begin{cases} \sigma((W_{i+1}^\top \delta_{i+1})_r + B_1) - B_1 = (W_{i+1}^\top \delta_{i+1})_r, & \sigma'(z_i)_r = 1, \\ \sigma((W_{i+1}^\top \delta_{i+1})_r - B_1) - 0 = 0, & \sigma'(z_i)_r = 0, \end{cases}$$

which equals $l(W_{i+1}^\top \delta_{i+1} r)_r \cdot \sigma'(z_i)_r$ in both cases. Therefore,

$$\mathrm{FFN}_3 \circ A^*(\begin{bmatrix} B_{i+1} \\ F_i \end{bmatrix}) = l((W_{i+1}^\top \delta_{i+1}) \odot \sigma'(z_i) r) \cdot 1_{1 \times d} = \delta_i \cdot 1_{1 \times d}.$$

**Step 3: Building the gradient block $\mathrm{vec}(\delta_i o_{i-1}^\top)$:** We next construct $\mathrm{vec}(\delta_i o_{i-1}^\top) \cdot 1_{1 \times d}$ from $\delta_i \cdot 1_{1 \times d}$ and $o_{i-1} \cdot 1_{1 \times d}$. Fix any $\epsilon_2 > 0$. By Lemma A.5, for each $h \in [d]$ there exists a multi-head attention module $\mathrm{Attn}_h$ such that

$$\| \mathrm{Attn}_h(\begin{bmatrix} \delta_i \cdot 1_{1 \times d} \\ o_{i-1} \cdot 1_{1 \times d} \end{bmatrix}) - \begin{bmatrix} 0_{(h+1)d \times d} \\ (o_{i-1})_h \delta_i \cdot 1_{1 \times d} \\ 0_{((d-h+1)d+1) \times d} \end{bmatrix} \|_\infty \leq \epsilon_2.$$

Summing over $h$ and using the triangle inequality yields

$$\| \sum_{h=1}^d \mathrm{Attn}_h(\begin{bmatrix} \delta_i \cdot 1_{1 \times d} \\ o_{i-1} \cdot 1_{1 \times d} \end{bmatrix}) - \sum_{h=1}^d \begin{bmatrix} 0_{(h+1)d \times d} \\ (o_{i-1})_h \delta_i \cdot 1_{1 \times d} \\ 0_{((d-h+1)d+1) \times d} \end{bmatrix} \|_\infty \leq d\epsilon_2.$$

To identify the deterministic sum with $\mathrm{vec}(\delta_i o_{i-1}^\top)$, recall the column-stacking convention

$$\mathrm{vec}(M) := (M_{1,1}, \ldots, M_{d,1}, M_{1,2}, \ldots, M_{d,2}, \ldots, M_{1,d}, \ldots, M_{d,d})^\top, \qquad M \in \mathbb{R}^{d \times d}.$$

Since

$$\delta_i o_{i-1}^\top = [(o_{i-1})_1 \delta_i, \ldots, (o_{i-1})_d \delta_i],$$

the $h$-th length-$d$ block of $\mathrm{vec}(\delta_i o_{i-1}^\top)$ equals $(o_{i-1})_h \delta_i$. Hence,

$$\sum_{h=1}^{d} \begin{bmatrix} 0_{(h+1)d \times d} \\ (o_{i-1})_h \delta_i \cdot 1_{1 \times d} \\ 0_{((d-h+1)d+1) \times d} \end{bmatrix} = \begin{bmatrix} 0_{2d \times d} \\ \mathrm{vec}(\delta_i o_{i-1}^\top) \cdot 1_{1 \times d} \\ 0_{(d+1) \times d} \end{bmatrix},$$

and therefore

$$\| \sum_{h=1}^{d} \mathrm{Attn}_h \left( \begin{bmatrix} \delta_i \cdot 1_{1 \times d} \\ o_{i-1} \cdot 1_{1 \times d} \end{bmatrix} \right) - \begin{bmatrix} 0_{2d \times d} \\ \mathrm{vec}(\delta_i o_{i-1}^\top) \cdot 1_{1 \times d} \\ 0_{(d+1) \times d} \end{bmatrix} \|_\infty \le d\epsilon_2.$$

**Step 4: Implement the gradient descent update on** $\mathrm{vec}(W_i)$**:** A transformer block outputs the sum of its branches, and token-wise linear maps can scale their outputs arbitrarily. Thus we implement the update in the $\mathrm{vec}(W_i)$-slot by summing: (i) a token-wise skip branch that places $\mathrm{vec}(W_i) \cdot 1_{1 \times d}$ into the $\mathrm{vec}(W_i)$ rows, and (ii) the attention branch $\sum_{h=1}^{d} \mathrm{Attn}_h(\cdot)$ from Step 3, scaled by $-\eta$. This yields (in the intended rows)

$$\mathrm{vec}(W_i) \cdot 1_{1 \times d} + l(-\eta \mathrm{vec}(\delta_i o_{i-1}^\top)r) \cdot 1_{1 \times d} = l(\mathrm{vec}(W_i) - \eta \mathrm{vec}(\delta_i o_{i-1}^\top)r) \cdot 1_{1 \times d}.$$

Note that we realize the factor $-\eta$ by absorbing it into the attention head's value, since $\mathrm{Attn}(Q, K, cV) = c\mathrm{Attn}(Q, K, V)$.

**Step 5: Compute** $W_i^\top \delta_i$ **for the next backward signal:** Fix any $\epsilon_3 > 0$. By Lemma A.5, for each $h \in [d]$ there exists a multi-head attention module $\mathrm{Attn}_h^*$ such that

$$\|\mathrm{Attn}_h^* \left( \begin{bmatrix} \delta_i \cdot 1_{1 \times d} \\ \mathrm{vec}((W_i)_{h,:}) \cdot 1_{1 \times d} \end{bmatrix} \right) - (W_i)_{h,:}^\top \delta_i \cdot e_{h+d}^{(3d+d^2+1)} \cdot 1_{1 \times d}\|_\infty \le \epsilon_3.$$

Summing over $h$ places $W_i^\top \delta_i \cdot 1_{1 \times d}$ into the designated coordinates (with total $l_\infty$ error at most $d\epsilon_3$).

**Step 6: Assemble a single transformer block and bound:** We now combine (a) the gradient term from Step 3 (scaled by $-\eta$), (b) the dot-product term from Step 5, and (c) a skip/placement branch that carries $\delta_i$ and $\mathrm{vec}(W_i)$. Define

$$\mathcal{F}_{\mathrm{backprop}}(Z) := -\eta \sum_{h=1}^{d} \mathrm{Attn}_h \left( \begin{bmatrix} \mathrm{FFN}_3 \circ A^*(Z) \\ EA^*(Z) \end{bmatrix} \right)$$
$$+ \sum_{h=1}^{d} \mathrm{Attn}_h^* \left( \begin{bmatrix} \mathrm{FFN}_3 \circ A^*(Z) \\ E_h A^*(Z) \end{bmatrix} \right) + \left( \begin{bmatrix} I_d \\ 0_{(2d+d^2+1) \times d} \end{bmatrix} \mathrm{FFN}_3 + A_s \right) \circ A^*(Z), \qquad (19)$$

where the token-wise selectors are

$$E := \begin{bmatrix} I_d & 0_{d \times (d^2+4d)} \end{bmatrix}, \qquad E_h := \begin{bmatrix} 0_{d \times (h+3)d} & I_d & 0_{d \times (d-h)d} \end{bmatrix}.$$

informing and the skip/placement map is

$$A_s(Z) := \begin{bmatrix} 0_{2d \times 4d} & 0_{2d \times d^2} \\ 0_{d^2 \times 4d} & I_{d^2} \\ 0_{(d+1) \times 4d} & 0_{(d+1) \times d^2} \end{bmatrix} Z.$$

Define token-wise linear padding maps $P_{\mathrm{top}} u := \begin{bmatrix} u \\ 0 \end{bmatrix}$ and $P_{\mathrm{bot}} v := \begin{bmatrix} 0 \\ v \end{bmatrix}$, so that $P_{\mathrm{top}}(\mathrm{FFN}_3(A^*(Z))) + P_{\mathrm{bot}}(EA^*(Z)) = \begin{bmatrix} \mathrm{FFN}_3(A^*(Z)) \\ EA^*(Z) \end{bmatrix}$. All maps $A^*, E, E_h, A_s, P_{\mathrm{top}}, P_{\mathrm{bot}}$ are fixed token-wise linear maps $(Z \mapsto TZ)$. Such that any $T$ can be absorbed into standard token wise projections. Now, for attention, replace $W_Q, W_K, W_V$ by $W_Q T, W_K T, W_V T$; for an FFN $\mathrm{FFN}(Z) = W_2 \sigma(W_1 Z)$, replace $W_1$ by $W_1 T$. Hence the terms $EA^*(Z)$ and $E_h A^*(Z)$ in (19) are realized by standard attention/FFN layers with modified projections, and $P_{\mathrm{top}}, P_{\mathrm{bot}}, A_s$ are standard token-wise padding/skip maps, so (19) uses only standard Transformer components.

Finally, the constructions in Steps 2–5 show that each required block of $B_i$ is produced (in the appropriate slots) up to errors $d\epsilon_2$ and $d\epsilon_3$ coming from Lemma A.5. Thus, for any target accuracy $\varepsilon > 0$, choosing $\epsilon_2 \leq \varepsilon/(2\eta d)$ and $\epsilon_3 \leq \varepsilon/(2d)$ gives

$$\|\mathcal{F}_{\text{backprop}}(\begin{bmatrix} B_{i+1} \\ F_i \end{bmatrix}) - B_i\|_\infty \leq \varepsilon,$$

except for an arbitrarily small input region corresponding to the transition interval of $\sigma'$. Since $\varepsilon > 0$ is arbitrary, this completes the proof. $\qquad\square$

## C.6 Proof of Proposition 4.1

For the special occasion when $X_l$ and $Y_l$ for $l \in [L]$ are given in the input, we also devise the following proposition to calculate the forward/ backward step for this input.

**Proposition C.1** (Proposition 4.1 Restated: Initialization of the First Forward Block)**.** *Fix $l \in [L]$ and let $X_l$, $B_1^{(l)}$, and $F_1^{(l)}$ be the blocks defined in Definition 4.3. Then for any $\varepsilon > 0$, there exists a single Transformer block (with residual/skip connections) $\mathcal{F}_x$ such that, for the input obtained by concatenating the two blocks,*

$$Z = \begin{bmatrix} B_1^{(l)} \\ X_l \end{bmatrix},$$

*the output $\mathcal{F}_x(Z)$ approximates the desired forward block $F_1^{(l)}$ such that*

$$l\|\mathcal{F}_x(Z) - F_1^{(l)}r\|_\infty \le \varepsilon.$$

*Proof.* Let $A_3$ be

$$A_3(Z) := \begin{bmatrix} \begin{bmatrix} 0_{d\times(1+3d+d^2)} & I_d & 0_{d\times(2d+d^2+1)} \end{bmatrix} Z \\ \begin{bmatrix} 0_{d^2\times 2d} & I_{d^2} & 0_{d^2\times(d+1)} \end{bmatrix} Z \end{bmatrix}.$$

This outputs

$$A_3\left(\begin{bmatrix} B_1^{(l)} \\ X_l \end{bmatrix}\right) = \begin{bmatrix} x_l \cdot 1_{1\times d} \\ \mathrm{vec}(W_1^{(l)}) \cdot 1_{1\times d} \end{bmatrix}.$$

Now according to Lemma A.5, for every $h \in [d]$, there is a multi-head attention $\mathrm{Attn}_h^{(2)}$ such that

$$\|\sum_{h=1}^{d} \mathrm{Attn}_h^{(2)}\left(\begin{bmatrix} x_l \cdot 1_{1\times d} \\ \mathrm{vec}(W_1^{(l)}) \cdot 1_{1\times d} \end{bmatrix}\right) - z_1^{(l)}\|_\infty \le \epsilon_4$$

for any $\epsilon_4 > 0$. This output then goes through a ReLU function and goes into the final output.

Then use a $\mathrm{FFN}_4$ constructed as $\mathrm{FFN}_1$ in Appendix C.4 satisfying

$$\mathrm{FFN}_4(z_1^{(l)}) = \sigma'(z_1^{(l)}).$$

Append $\mathrm{FFN}_4$ to the output of $\sum_{h=1}^{d} \mathrm{Attn}_h^{(2)}$ outputs $\sigma'(z_1^{(l)})1_{1\times d}$.

Concatenating the outputs of $\sum_{h=1}^{d} \mathrm{Attn}_h^{(2)}$ and $\mathrm{FFN}_4$ with $x_l 1_{1\times d}$ and $\mathrm{vec}(W_1^{(l)}) \cdot 1_{1\times d}$ in the input yields

$$\begin{bmatrix} \sigma(\sum_{h=1}^{d} \mathrm{Attn}_h^{(2)}\left(\begin{bmatrix} x_l \cdot 1_{1\times d} \\ \mathrm{vec}(W_1^{(l)}) \cdot 1_{1\times d} \end{bmatrix}\right)) \\ x_l \cdot 1_{1\times d} \\ \sigma'(z_1^{(l)}) \cdot 1_{1\times d} \\ \mathrm{vec}(W_1^{(l)}) \cdot 1_{1\times d} \\ 1_{1\times d,} \end{bmatrix}.$$

That's closer to $F_1^{(l)}$ by $\epsilon_4$ in infinite norm. $\qquad\square$

## C.7 Proof of Proposition 4.2

**Proposition C.2** (Proposition 4.2 Restated: Initialization of the Backward Block)**.** *Fix $l \in [L]$, and let $F_N^{(l)}$, $Y_l$, and $B_N^{(l+1)}$ be the blocks defined in Definition 4.3. Then for any $\epsilon > 0$, there exists a single Transformer block (with residual/skip*

*connections) $\mathcal{F}_y$ such that, for the input obtained by concatenating the two blocks,*

$$Z = \begin{bmatrix} F_N^{(l)} \\ Y_l \end{bmatrix},$$

*the output $\mathcal{F}_y(Z)$ approximates the desired backward block $B_N^{(l+1)}$ such that*

$$\|\mathcal{F}_y(Z) - B_N^{(l+1)}\|_\infty \leq \epsilon.$$

*Proof.* First we use a token-wise linear transformation $A_4$ to extract the needed rows from the input.

$$A_4\left(\begin{bmatrix} F_N^{(l)} \\ Y_l \end{bmatrix}\right) := \begin{bmatrix} o_N^{(l)} \cdot 1_{1\times d} \\ y_l \cdot 1_{1\times d} \\ \text{vec}(W_N^{(l)}) \cdot 1_{1\times d} \end{bmatrix}.$$

Then, there exists a feed-forward network $\text{FFN}_5$ such that

$$\text{FFN}_5(o_N^{(l)}) = \sigma'(o_N^{(l)}) = \sigma'(z_N^{(l)})$$

except for an arbitrarily small region.

Same as in Appendix C.5 by Lemma A.5, there is a set of $d$ multi-head attention, whose sum we note by a larger multi-head attention $\text{Attn}_m^{(1)}$ that satisfies

$$\|\text{Attn}_m^{(1)}\left(\begin{bmatrix} o_N^{(l)} \cdot 1_{1\times d} \\ y_l \cdot 1_{1\times d} \\ \text{vec}(W_N^{(l)}) \cdot 1_{1\times d} \end{bmatrix}\right) - \text{vec}(\delta_N^{(l+1)}(o_N^{(l)})^\top) \cdot 1_{1\times d}\|_\infty \leq \epsilon_5$$

for any $\epsilon_5 > 0$.

Finally, upon the output of $\text{FFN}_5$, and choosing the loss to be squared loss, there is an $\text{FFN}_6$ that calculates

$$\underbrace{2(y_l - o_N^{(l)}) \odot \sigma'(z_N^{(l)})}_{\delta_N^{(l+1)}}.$$

Concatenating the above result with token-wise linear transformations yields

$$\begin{bmatrix} \delta_N^{(l+1)} \cdot 1_{1\times d} \\ (W_N^{(l)})^\top \delta_N^{(l+1)} \cdot 1_{1\times d} \\ \text{vec}(W_N^{(l)}) - \eta \text{Attn}_m^{(1)}\left(\begin{bmatrix} o_N^{(l)} \cdot 1_{1\times d} \\ y_l \cdot 1_{1\times d} \\ \text{vec}(W_N^{(l)}) \cdot 1_{1\times d} \end{bmatrix} \cdot 1_{1\times d}\right) \\ 0_{(d+1)\times d} \end{bmatrix},$$

whose difference with $B_N^{(l+1)}$ is bounded by $\epsilon_5$. $\qquad\square$

## C.8 Proof of Proposition 4.3

**Proposition C.3** (Proposition 4.3 Restated: Universal One-Step Update Block). *Fix $\varepsilon > 0$. There exists a single-layer Transformer block* $\mathrm{TF} : \mathbb{R}^{D \times d} \to \mathbb{R}^{D \times d}$ *with the following property.*

*Let $\mathcal{Z}$ denote the set of valid routed inputs $Z \in \mathbb{R}^{D \times d}$ that arise from the block formats in Definition 4.3, i.e., $Z$ is exactly one of the four types:* forward-type, backward-type, $x$-type, *or $y$-type. Let $M > 0$ be any constant such that $\|Z\|_\infty \leq M/2$ for all $Z \in \mathcal{Z}$.*

*Then for every $Z \in \mathcal{Z}$,*

$$\|\mathrm{TF}(Z) - T(Z)\|_\infty \leq \varepsilon,$$

*where $T$ is the corresponding target one-step update map:*

$$T(Z) = \begin{cases} T_{\mathrm{f}}(Z), & Z \text{ is forward-type}, \\ T_{\mathrm{b}}(Z), & Z \text{ is backward-type}, \\ T_x(Z), & Z \text{ is } x\text{-type}, \\ T_y(Z), & Z \text{ is } y\text{-type}. \end{cases}$$

*Proof.* **Goal.** Construct a *single* Transformer block that, on any valid routed input block $Z$, selects the correct sub-computation (forward, backward, $x$ update, or $y$ update) and returns the corresponding next block.

**Roadmap.** *Step 1:* Build the exact type selectors from two designated routing rows. *Step 2:* Execute the four branch computations in parallel inside one enlarged attention/FFN block. *Step 3:* Gate the four candidate outputs and sum; since exactly one selector is active, the sum equals the one step update that we want.

**Step 1: Exact selectors from routing rows.** Let $Z \in \mathbb{R}^{D \times d}$ be a valid routed input block, and let

$$r_u := d^2 + 3d + 1, \qquad r_v := 2d^2 + 6d + 2,$$

with row extractors

$$u(Z) := Z_{r_u,:} \in \mathbb{R}^{1 \times d}, \qquad v(Z) := Z_{r_v,:} \in \mathbb{R}^{1 \times d}.$$

By the block design of $F_i, B_i, X_s, Y_s$, every $Z$ satisfies exactly one of the tag identities (entrywise across the $d$ columns):

$$\begin{array}{lll} \text{(forward-type)} & u(Z) - v(Z) = \mathbf{1}_{1 \times d}, \\ \text{(backward-type)} & -u(Z) - v(Z) + \mathbf{1}_{1 \times d} = 0, \\ (x\text{-type}) & -u(Z) + v(Z) = \mathbf{1}_{1 \times d}, \\ (y\text{-type}) & u(Z) + v(Z) = 3\mathbf{1}_{1 \times d}. \end{array}$$

Define the affine tests

$$\begin{aligned} a_f(Z) &:= u(Z) - v(Z) - \mathbf{1}_{1 \times d}, \\ a_b(Z) &:= -u(Z) - v(Z) + \mathbf{1}_{1 \times d}, \\ a_x(Z) &:= -u(Z) + v(Z) - \mathbf{1}_{1 \times d}, \\ a_y(Z) &:= u(Z) + v(Z) - 3\mathbf{1}_{1 \times d}. \end{aligned}$$

By the block design in Definition 4.3, these tests separate the four valid routed input types: for the unique matching type, the corresponding test equals 0, while the other three are bounded away from 0. Since each $a$ is affine, we may scale them by a fixed constant so that for every valid routed input, the non-matching tests satisfy

$$|a(Z)| \geq \mathbf{1}_{1 \times d}$$

entrywise across the $d$ columns.

Now define the selector function

$$\chi(w) := \mathrm{ReLU}(w+1) - 2\mathrm{ReLU}(w) + \mathrm{ReLU}(w-1),$$

applied entrywise. This satisfies

$$\chi(0) = 1, \qquad \chi(w) = 0 \quad \text{for all } |w| \geq 1.$$

Set

$$\gamma_f(Z) := \chi(a_f(Z)), \quad \gamma_b(Z) := \chi(a_b(Z)), \quad \gamma_x(Z) := \chi(a_x(Z)), \quad \gamma_y(Z) := \chi(a_y(Z)).$$

Hence, for every $Z \in \mathcal{Z}$, exactly one of the four selector rows equals $\mathbf{1}_{1\times d}$ and the other three equal $0_{1\times d}$.

**Step 2: Parallel candidate updates inside one block.** Fix $\varepsilon > 0$. Let

$$\mathcal{F}_{\mathrm{fwd}}, \quad \mathcal{F}_{\mathrm{bwd}}, \quad \mathcal{F}_x, \quad \mathcal{F}_y$$

denote the single-block constructions from Theorem 4.1, Theorem 4.2, Proposition 4.1, and Proposition 4.2, respectively, each chosen to $\varepsilon$-approximate its corresponding target map in $\|\cdot\|_\infty$.

Each of these four constructions consists of: (i) a finite family of attention heads, (ii) a token-wise FFN, and (iii) token-wise linear skip/placement maps. By block-diagonalizing the $Q/K/V$ projections and concatenating all heads, we may execute the four attention families in parallel inside one attention sublayer, writing their outputs into disjoint coordinate slices of a widened hidden state. Likewise, by closure of token-wise FFNs under parallelization and affine postcomposition, one widened FFN can simultaneously realize the four branch-specific post-processing maps in disjoint slices.

Thus, after one attention sublayer followed by one widened FFN, we obtain four candidate outputs

$$\widehat{T}_f(Z), \qquad \widehat{T}_b(Z), \qquad \widehat{T}_x(Z), \qquad \widehat{T}_y(Z) \in \mathbb{R}^{D\times d},$$

such that, outside the corresponding bad regions,

$$\|\widehat{T}_f(Z) - T_f(Z)\|_\infty \leq \varepsilon, \qquad \|\widehat{T}_b(Z) - T_b(Z)\|_\infty \leq \varepsilon,$$

$$\|\widehat{T}_x(Z) - T_x(Z)\|_\infty \leq \varepsilon, \qquad \|\widehat{T}_y(Z) - T_y(Z)\|_\infty \leq \varepsilon.$$

**Step 3: Gate and aggregate the candidates.** Since each target output is itself a valid block, its entries are bounded. Fix $M > 0$ so that

$$\|T_f(Z)\|_\infty, \|T_b(Z)\|_\infty, \|T_x(Z)\|_\infty, \|T_y(Z)\|_\infty \leq \frac{M}{2} \qquad \text{for all } Z \in \mathcal{Z},$$

and enlarge $M$ if we need to such that

$$\|\widehat{T}_f(Z)\|_\infty, \|\widehat{T}_b(Z)\|_\infty, \|\widehat{T}_x(Z)\|_\infty, \|\widehat{T}_y(Z)\|_\infty \leq M$$

on the valid region.

For a bounded block $Y \in \mathbb{R}^{D\times d}$ and a selector row $\gamma \in \{0,1\}^{1\times d}$, define the broadcast gate

$$G(Y, \gamma) := \mathrm{ReLU}(Y - M(\mathbf{1}_{D\times d} - \mathbf{1}_{D\times 1}\gamma)) - \mathrm{ReLU}(-Y - M(\mathbf{1}_{D\times d} - \mathbf{1}_{D\times 1}\gamma)),$$

applied entrywise. If $\|Y\|_\infty \leq M$, then

$$G(Y, \mathbf{1}_{1\times d}) = Y, \qquad G(Y, 0_{1\times d}) = 0_{D\times d}.$$

Therefore the same widened FFN may output

$$\mathrm{TF}(Z) := G(\widehat{T}_f(Z), \gamma_f(Z)) + G(\widehat{T}_b(Z), \gamma_b(Z)) + G(\widehat{T}_x(Z), \gamma_x(Z)) + G(\widehat{T}_y(Z), \gamma_y(Z)).$$

Now fix any valid routed input $Z \in \mathcal{Z}$. By definition of the target map,

$$T(Z) = \begin{cases} T_f(Z), & Z \text{ is forward-type}, \\ T_b(Z), & Z \text{ is backward-type}, \\ T_x(Z), & Z \text{ is } x\text{-type}, \\ T_y(Z), & Z \text{ is } y\text{-type}. \end{cases}$$

Since exactly one selector is active, the gated sum defining $\mathrm{TF}(Z)$ reduces to the candidate output associated with the type of $Z$. Thus,

$$\mathrm{TF}(Z) = \begin{cases} \widehat{T}_f(Z), & Z \text{ is forward-type}, \\ \widehat{T}_b(Z), & Z \text{ is backward-type}, \\ \widehat{T}_x(Z), & Z \text{ is } x\text{-type}, \\ \widehat{T}_y(Z), & Z \text{ is } y\text{-type}. \end{cases}$$

Let $\mathcal{B}$ denote the union of the bad regions associated with the four branch constructions. Then for every $Z \in \mathcal{Z} \setminus \mathcal{B}$,

$$\|\mathrm{TF}(Z) - T(Z)\|_\infty \le \varepsilon.$$

For example, if $Z$ is forward type, then

$$\|\mathrm{TF}(Z) - T(Z)\|_\infty = \|\widehat{T}_f(Z) - T_f(Z)\|_\infty \le \varepsilon,$$

and the backward, $x$, and $y$ type cases are identical.

Finally, the construction above uses exactly one attention sublayer (all branch heads in parallel) and one token wise FFN (branch specific post processing, selector construction, gating, and aggregation), together with standard token-wise linear skip/placement maps. Therefore TF is a single-layer Transformer block, as claimed. $\qquad\square$

## C.9 Proof of Proposition 4.4

**Proposition C.4** (In-context GD State Trajectory via Single Transformer Block). *Fix integers $N \geq 1$ and $L \geq 1$, and a labeled sequence $(x_l, y_l)_{l=1}^{L}$ with input formatting as above. Let $\varepsilon_{C1}, \varepsilon_{4.1}, \varepsilon_{C2}, \varepsilon_{4.2} > 0$ be the accuracy parameters from Proposition 4.1, Theorem 4.1, Proposition 4.2, and Theorem 4.2, and let $\varepsilon_{C3} > 0$ be the routing accuracy parameter from Proposition 4.3. Define*

$$\varepsilon_{\text{round}} := \varepsilon_{C1} + (N-1)\varepsilon_{4.1} + \varepsilon_{C2} + (N-1)\varepsilon_{4.2} + (2N+1)\varepsilon_{C3}.$$

*Assume that the exact round update map is $K$-Lipschitz in $\|\cdot\|_{\infty,\max}$. Then there exists a single-layer Transformer block* TF *defined in Proposition 4.3, such that it produces state blocks $\{\widehat{B}_i^{(l)}\}_{i\in[N],l=1}^{L+1}$ satisfying $\widehat{B}_i^{(1)} = B_i^{(1)}$ and, for every $l \in [L]$,*

$$\max_{i\in[N]} \|\widehat{B}_i^{(l+1)} - B_i^{(l+1)}\|_{\infty} \leq \varepsilon_{\text{round}} \sum_{t=0}^{l-1} K^t.$$

*In particular,*

$$\max_{i\in[N]} \|\widehat{B}_i^{(L+1)} - B_i^{(L+1)}\|_{\infty} \leq \varepsilon_{\text{round}} \sum_{t=0}^{L-1} K^t.$$

*Proof.* **Proof outline.** We prove the claimed $L$-round state error bound in three steps: (i) show that, for a *fixed* round $l$, one full forward–backward sweep implemented by TF incurs at most $\varepsilon_{\text{round}}$ additional $\|\cdot\|_{\infty}$ error in the resulting state blocks; (ii) formalize the stability of the update map (iii) combine (i)–(ii) to obtain the recursion $E_{l+1} \leq E_l + \varepsilon_{\text{round}}$ and unroll it.

**Step 0 (norm and notation).** For a stack of state blocks $B^{(l)} := [B_1^{(l)} \cdots B_N^{(l)}]$, define

$$\|B^{(l)}\|_{\infty,\max} := \max_{i\in[N]} \|B_i^{(l)}\|_{\infty}, \qquad E_l := \|\widehat{B}^{(l)} - B^{(l)}\|_{\infty,\max}.$$

By exact initialization, $E_1 = 0$.

**Step 1 (one-round approximation error is $\leq \varepsilon_{\text{round}}$).** Fix an arbitrary round $l \in [L]$. Let $\mathcal{U}_{\ell}$ denote the exact round-$l$ update map $[B_1^{(l)}, \ldots, B_N^{(l)}] \mapsto [B_1^{(l+1)}, \ldots, B_N^{(l+1)}]$, and let $\widehat{\mathcal{U}}_l$ be the corresponding map implemented by TF with routing.

By Proposition C.3, each invocation of TF on an input executes the intended sub computation (Prop. C.1 / Thm. 4.1 / Prop. C.2 / Thm. 4.2) with an additional $\|\cdot\|_{\infty}$ error at most $\varepsilon_{C3}$. Combining this routing error with the approximation errors $\varepsilon_{C1}, \varepsilon_{4.1}, \varepsilon_{C2}, \varepsilon_{4.2}$ accrued across the $N$ forward steps and $N$ backward steps yields the per-round bound

$$\|\widehat{\mathcal{U}}_l(B^{(l)}) - \mathcal{U}_l(B^{(l)})\|_{\infty,\max} \leq \varepsilon_{\text{round}},$$

where

$$\varepsilon_{\text{round}} = \varepsilon_{C1} + (N-1)\varepsilon_{4.1} + \varepsilon_{C2} + (N-1)\varepsilon_{4.2} + (2N+1)\varepsilon_{C3}.$$

**Step 2 (stability of the exact update map).** Assume there exists a constant $K > 0$ such that for every $l \in [L]$ and any two admissible state stacks $B, B'$,

$$\|\mathcal{U}_l(B) - \mathcal{U}_l(B')\|_{\infty,\max} \leq K \|B - B'\|_{\infty,\max}.$$

**Step 3 (one-step recursion).** Now, notice that

$$\begin{aligned}
E_{l+1} &= \|\widehat{B}^{(l+1)} - B^{(l+1)}\|_{\infty,\max} \\
&= \|\widehat{\mathcal{U}}_l(\widehat{B}^{(l)}) - \mathcal{U}_l(B^{(l)})\|_{\infty,\max} \\
&\leq \|\widehat{\mathcal{U}}_l(\widehat{B}^{(l)}) - \mathcal{U}_l(\widehat{B}^{(l)})\|_{\infty,\max} + \|\mathcal{U}_l(\widehat{B}^{(l)}) - \mathcal{U}_l(B^{(l)})\|_{\infty,\max} \quad \text{(triangle inequality)} \\
&\leq \varepsilon_{\text{round}} + K_{\text{stab}} \|\widehat{B}^{(l)} - B^{(l)}\|_{\infty,\max} \quad \text{(Step 1 + Step 2)} \\
&= \varepsilon_{\text{round}} + K E_l.
\end{aligned}$$

Unrolling this recursion using $E_1 = 0$ yields

$$E_{l+1} \leq \varepsilon_{\text{round}} \sum_{t=0}^{l-1} K^t \qquad \text{for all } l \in [L].$$

Taking $l = L$ gives

$$E_{L+1} \leq \varepsilon_{\text{round}} \sum_{t=0}^{L-1} K^t.$$

$\square$

# D  Initialization and Dynamic Masking

In this appendix, we explain how the initialization cases fit together with the dynamic masking rule in Definition 4.4. Recall that the initial input format is

$$Z^{(1)} = [X_1, Y_1, \ldots, X_L, Y_L, B_1^{(1)}, \ldots, B_N^{(1)}].$$

Here $Z^{(1)}$ denotes the initial prompt before CoT generation begins. The generation index $t$ counts only blocks generated after this initial prompt; thus $t = 1$ is the first generated CoT block.

There are three kinds of steps.

**Forward Initialization Step.**  At the start of round $\ell$, the first forward block is generated by Proposition 4.1:

$$F_1^{(\ell)} arrow [B_1^{(\ell)}, X_\ell].$$

This starts the forward sweep for the data point $(x_\ell, y_\ell)$.

**Backward Initialization Step.**  After the forward sweep reaches $F_N^{(\ell)}$, the top backward block is generated by Proposition 4.2:

$$B_N^{(\ell+1)} arrow [F_N^{(\ell)}, Y_\ell].$$

This starts the backward sweep.

**Interior Recurrent Steps.**  All remaining forward and backward blocks are generated by the dynamic masking rule from Definition 4.4:

$$\mathcal{M}(t) = (t - 1, t - 2((t - 1) \bmod N) - 1).$$

This rule retrieves the previous generated block and the matching block from the opposite sweep.

For a forward interior step, this gives

$$F_i^{(\ell)} arrow [F_{i-1}^{(\ell)}, B_i^{(\ell)}], \qquad i = 2, \ldots, N.$$

For a backward interior step, this gives

$$B_i^{(\ell+1)} arrow [B_{i+1}^{(\ell+1)}, F_i^{(\ell)}], \qquad i = N - 1, \ldots, 1.$$

The only special case is the first forward sweep. In this case, $B_2^{(1)}, \ldots, B_N^{(1)}$ have not been generated by the CoT trajectory; they are already supplied in the initial prompt. Therefore, in the first round we explicitly use

$$F_i^{(1)} arrow [F_{i-1}^{(1)}, B_i^{(1)}], \qquad i = 2, \ldots, N.$$

This corresponds to the branch

$$[F_{t-1}^{(1)}, B_t^{(1)}], \qquad 2 \le t \le N,$$

which is only needed to begin the first forward sweep.

**Example: $N = 3$.**  For $N = 3$, the initial input is

$$Z^{(1)} = [X_1, Y_1, \ldots, X_L, Y_L, B_1^{(1)}, B_2^{(1)}, B_3^{(1)}].$$

The generated trajectory begins as follows:

| $t$ | generated block | retrieved pair | case |
|---|---|---|---|
| 1 | $F_1^{(1)}$ | $[B_1^{(1)}, X_1]$ | Proposition 4.1 |
| 2 | $F_2^{(1)}$ | $[F_1^{(1)}, B_2^{(1)}]$ | first forward sweep |
| 3 | $F_3^{(1)}$ | $[F_2^{(1)}, B_3^{(1)}]$ | first forward sweep |
| 4 | $B_3^{(2)}$ | $[F_3^{(1)}, Y_1]$ | Proposition 4.2 |
| 5 | $B_2^{(2)}$ | $[B_3^{(2)}, F_2^{(1)}]$ | mod rule |
| 6 | $B_1^{(2)}$ | $[B_2^{(2)}, F_1^{(1)}]$ | mod rule |
| 7 | $F_1^{(2)}$ | $[B_1^{(2)}, X_2]$ | Proposition 4.1 |
| 8 | $F_2^{(2)}$ | $[F_1^{(2)}, B_2^{(2)}]$ | mod rule |
| 9 | $F_3^{(2)}$ | $[F_2^{(2)}, B_3^{(2)}]$ | mod rule |
| 10 | $B_3^{(3)}$ | $[F_3^{(2)}, Y_2]$ | Proposition 4.2 |

For example, at $t = 8$,

$$\mathcal{M}(8) = (8 - 1, 8 - 2((8 - 1) \bmod 3) - 1) = (7, 5).$$

The seventh generated block is $F_1^{(2)}$, and the fifth generated block is $B_2^{(2)}$. Hence the retrieved pair is

$$[F_1^{(2)}, B_2^{(2)}],$$

which is exactly the input needed to generate $F_2^{(2)}$.

This also explains why the first forward sweep is separated. At $t = 2$, the mod rule would give

$$\mathcal{M}(2) = (1, -1),$$

which is not a valid generated-block index. The missing block is $B_2^{(1)}$, which belongs to the initial prompt rather than the generated trajectory. Therefore the first forward sweep is handled explicitly, while the mod rule governs the recurrent interior trajectory.

# E  Experimental Studies

**Objectives.**  We provide three proof of concept numerical experiments to support our theoretical results.

- **Statistical Model in Section 3.** The first experiment studies the statistical models in Section 3. Its goal is to verify that a single shared transformer learn the exact one-step optimization update for generalized linear models, ridge regression, and lasso regression.

- **Forward (Theorem 4.1) and Backward (Theorem 4.2) Transformer Blocks.** The second experiment studies the operator level construction in Section 4. Its goal is to verify that the forward and backward transformer blocks learn the local update rules induced by dynamic masking when they are trained separately.

- **End-to-End Training for ICGD of Feed-Forward Networks.** The third experiment studied end-to-end ICGD for feed-forward networks. Its goal is to verify that one shared transformer autoregressively generate the full one round CoT trajectory, so that the forward and backward updates are learned jointly rather than by two separate models. We further test whether the same trained transformer can be reused across multiple rounds, where the rollout error exhibits the accumulation behavior proposed by Proposition 4.4

## E.1  ICGD of Statistical Models

**Model Architecture.**  For each statistical-model family in Section 3 (GLM, Ridge, and Lasso) we train a separate decoder-only transformer with the same architecture. For each example, the prompt contains the observed dataset together with an initialization $w^{(1)} \in \mathbb{R}^d$, and the model autoregressively predicts the two-token trajectory

$$\big(g^{(1)}, \, w^{(2)}\big),$$

where $g^{(1)}$ is the exact one-step gradient token and $w^{(2)}$ is the exact parameter after one optimization step.

**Data Generation.**  We follow the setups in Section 3 for GLM, ridge, and lasso. In all three cases, we sample a teacher parameter $w^\star \in \mathbb{R}^d$ and an independent initialization $w^{(1)} \in \mathbb{R}^d$ from a Gaussian distribution, and sample covariates

$$x_i \overset{\text{i.i.d.}}{\sim} \text{Unif}([-1, 1]^d).$$

For GLM, we generate responses as

$$y_i = \sigma\big((w^\star)^\top x_i\big),$$

and compute the exact one-step gradient-descent update using the GLM objective from Section 3 with step size $\eta = 0.1$.

For ridge and lasso, we generate

$$y_i = (w^\star)^\top x_i + \varepsilon_i, \qquad \varepsilon_i \overset{\text{i.i.d.}}{\sim} \mathcal{N}(0, \sigma^2), \qquad \sigma = 0.05.$$

We then compute the exact one-step update using the ridge objective from Section 3 with $\lambda_{\text{ridge}} = 0.1$ and $\eta = 0.1$, and the lasso proximal-gradient update from Section 3 with $\lambda_{\text{lasso}} = 0.08$ and $\eta = 0.1$.

Thus, in each setting, the target trajectory is

$$\big(g^{(1)}, \, w^{(2)}\big),$$

where $g^{(1)}$ is the exact gradient token at $w^{(1)}$, and $w^{(2)}$ is obtained by one exact gradient step for GLM and ridge, and by one exact proximal-gradient step for lasso.

**Training Objective.**  For each statistical-model setting, we train the transformer autoregressively on the exact two-token trajectory. If

$$\big(\widehat{g}^{(1)}, \, \widehat{w}^{(2)}\big)$$

denotes the model prediction from the prompt $(X, y, w^{(1)})$, then the training loss is

$$\mathcal{L}_{\text{traj}} = \|\widehat{g}^{(1)} - g^{(1)}\|_2^2 + \|\widehat{w}^{(2)} - w^{(2)}\|_2^2.$$

**Evaluation and Metrics.** For each setting, we evaluate the trained model in two autoregressive modes. In teacher-forced evaluation, the second prediction is conditioned on the true first token $g^{(1)}$. In free-running evaluation, the second prediction is conditioned on the model's own first prediction $\widehat{g}^{(1)}$.

We report the trajectory MSE in both settings:

$$\text{MSE}_{\text{TeacherForced}} = \frac{1}{2}\left(\|\widehat{g}^{(1)} - g^{(1)}\|_2^2 + \|\widehat{w}^{(2)}_{\text{TeacherForced}} - w^{(2)}\|_2^2\right),$$

and

$$\text{MSE}_{\text{FreeRunning}} = \frac{1}{2}\left(\|\widehat{g}^{(1)} - g^{(1)}\|_2^2 + \|\widehat{w}^{(2)}_{\text{FreeRunning}} - w^{(2)}\|_2^2\right).$$

All values are averaged over 10 random seeds; in the figure, black markers and error bars denote the sample mean and standard deviation.

**Results.** Across GLM, ridge, and lasso, the free-running MSE remains close to the teacher-forced MSE. This shows that the autoregressive transformer accurately learns the corresponding one-step optimization map in each statistical-model setting.

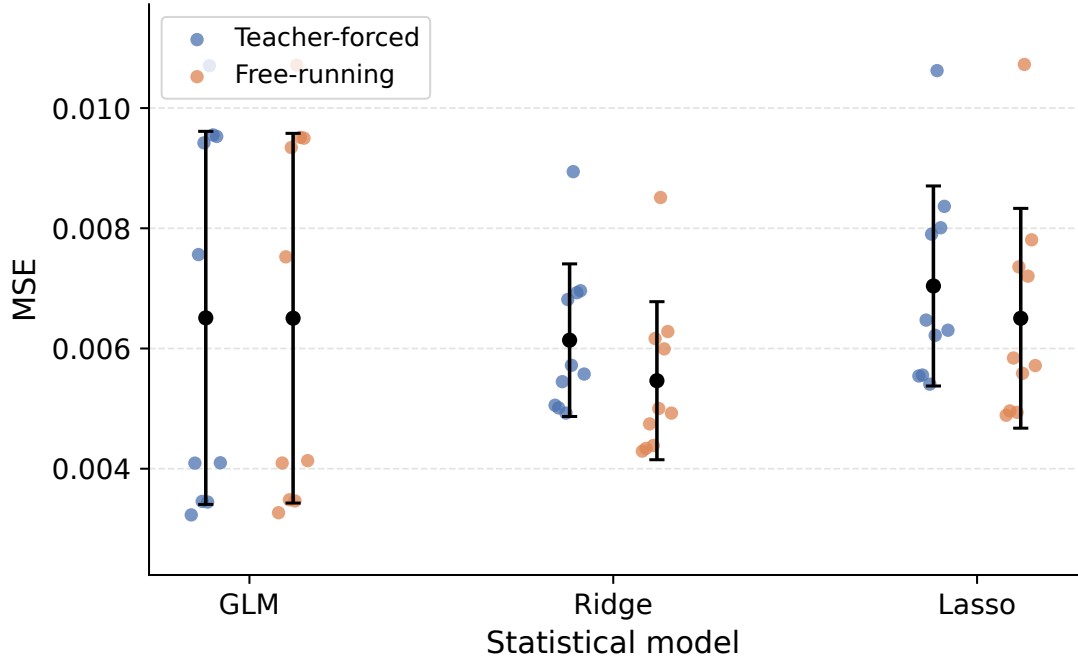

*Figure 2.* **ICGD of Statistical Models.** We consider three one-step settings: GLM, ridge, and lasso. In each case, we sample a teacher parameter $w^\star \in \mathbb{R}^d$ and a separate student initialization $w_1$. For GLM, we sample $x$ from $\text{Uniform}[-1, 1]$ and set $y = \sigma(x^\top w^\star)$. For ridge and lasso, we sample $X$ from $\text{Uniform}[-1, 1]$ and set $y = x^\top w^\star + \varepsilon$ with $\varepsilon \sim N(0, \sigma^2 I)$. The transformer input is $[X, Y, w_1]$. The target trajectory is $(\nabla\mathcal{L}(w^{(l)}), w^{(l+1)})$, where $\nabla\mathcal{L}(w^{(l)})$ is the exact one-step gradient at $w^1$, and $w^{(l+1)}$ is the parameter obtained after one exact update step from $w^{(1)}$; for GLM and ridge this is one gradient-descent step, while for lasso it is one proximal-gradient step. Teacher-forced uses the true previous token to predict the next token. Free-running uses the model's own previous output. The figure reports the resulting MSE over 10 seeds; black markers and bars denote the mean and standard deviation. In all three settings, free-running remains close to teacher-forced, showing that one shared transformer learns the one-step statistical-model update jointly.

### E.2 Learning Separate Forward and Backward ICGD Operators

**Objectives.** The goal of the experiment section is to demonstrate single single-layer decoder-only transformer can simulate multi-step ICGD process for $N$-layer neural networks by generating CoT sequence.

We aim to train a single-layer autoregressive decoder-only transformer, augmented with the proposed dynamic masking, to generate the CoT sequence that simulates the multi-step ICGD process for a variety of $N$-layer neural network.

In first step we train the forward operator and backward operator seperately. For forward operator, the input data is in the form of Theorem 4.1. Formally, the training dataset for forward transformer block $\mathcal{F}_{\text{forward}}$ for $k$-th teacher neural network is $\mathcal{X}^{k,\text{forward}} := \{X_i^{k,\text{forward}}, Y_i^{k,\text{forward}}\}_{1 \leq i \leq N}$ where each data pairs is

$$X_i^{k,\text{forward}} = \begin{bmatrix} F_{i-1}^k \\ B_i^k \end{bmatrix}, \quad Y_i^{k,\text{forward}} = F_i^k.$$

We hope by this training data, once the forward transformer block $\mathcal{F}_{\text{forward}}$ see the input selected by dynamic masking, it output the desired next-step forward blocks. In our theory, the transformer blocks can perform the same forward calculation no matter which neural network we want to simulate its forward calculation path. Hence in the training data, we also create $K$ different neural networks training data. That is $\mathcal{X}^{\text{forward}} = \{\mathcal{X}^{k,\text{forward}}\}_{1 \leq k \leq K}$.

Similarly, for backward operator, we form the training dataset as in Theorem 4.2 and the above logic to train a backward operator transformer blocks $\mathcal{F}_{\text{backprop}}$. $\mathcal{X}^{k,\text{backprop}} := \{X_i^{\text{backprop}}, Y_i^{\text{backprop}}\}_{1 \leq i \leq N}$ where each datapoint is

$$X_i^{k,\text{backprop}} = \begin{bmatrix} B_{i+1}^k \\ F_i^k \end{bmatrix}, \quad Y_i^{k,\text{backprop}} = B_i^k.$$

For simplicity, we set $d = 5$, start from $N = 3$, and set total GD updates step to be $L = 20$.

**Model Architecture.** We use a one-layer transformer block, containing one multi-head attention and one FFN layer. We set the output dimension of the FFN layer to match the dimension of column vector in $F_i$. The number of head we test starting from $h = 8$.

**Data Generation.** We set the hidden dimension of the simulated neural network with $d \in \{10, 20\}$, and the number of layer $N \in \{3, 6, 9\}$.

**Training Objective.** During training we sample mini batches that mix all teachers' neural networks $k \in [K]$, GD rounds $\ell \in [L]$, and hidden layers $i \in [N]$. Given a batch of $B$ pairs $(X_b, Y_b)$, where each $X_b \in \mathbb{R}^{d \times R}$ and $Y_b \in \mathbb{R}^{d \times R}$ (with $R = 1 + 3d + d^2$), we minimise the mean squared Frobenius error

$$\mathcal{L}_{\text{forward}} = \frac{1}{B} \sum_{b=1}^{B} \|\mathcal{F}_{\text{forward}}(X_b^{\text{forward}}) - Y_b^{\text{forward}}\|_F^2.$$

We use the same train loss for the backward operator. Thus, every parameter update encourages the model to reproduce all entries of the next block, and the random batching forces the transformers to generalize across different layers $i$, optimization steps $\ell$, and teacher networks $k$.

**Evaluation and Metrics.** We evaluate whether the trained single-layer transformer blocks ($\mathcal{F}_{\text{forward}}$ and $\mathcal{F}_{\text{backprop}}$) simulate a full $2NL$-step ICGD trajectory. Specifically, we initialize with the input-output pair $(X_s, Y_s)$, and then repeatedly apply these two learned operators (by using dynamic masking to select appropriate prior blocks, $[F_{i-1}, B_i]$ for foward, $[B_{i+1}, F_i]$ for backward) to generate forward and backward blocks, forming a complete chain-of-thought sequence of length $2NL$. At each backward block $B_i$ (particularly at steps $2N, 4N, 6N, \ldots, 2NL$), we extract the flattened weight matrix $W^l$ (rows $2d+1$ to $2d+d^2$ of $B_i$) from the generated block and measure the squared Frobenius norm of deviation from the ground-truth teacher weights

$$\|W_{\text{CoT}}^l - W_{\text{true}}^l\|_F^2.$$

**Result.** We repeated the experiment for 5 rounds (each round with a different set of training and validation data) and in each of them we observed that in just 20 epochs of training, the network has successfully learned how to accomplish the

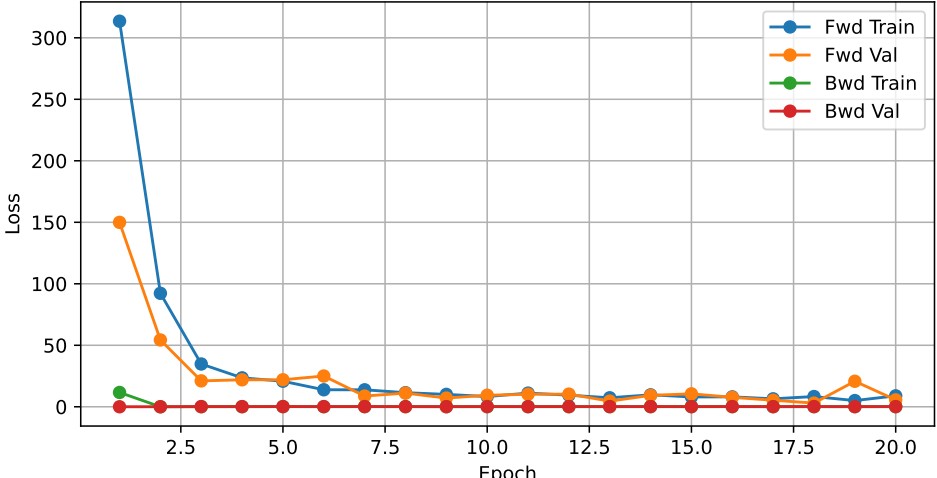

*Figure 3.* The result of our experiment, four lines denote training loss and validation loss for simulated forward propagation and backward propagation.

forward propagation and the backward propagation. We show the result of our training in Figure 3. The minimal validation losses for forward and backward processes are 3.00 and 0.139.

### E.3   End-to-End Training of ICGD for FFN

**Model Architecture.**   For each random seed, we train one decoder-only transformer to autoregressively predict the full one-round ICGD trajectory of a 3-layer FFN. The prompt consists of $(x, y)$ together with the initial student layer states. The target continuation is

$$\left(F_0^{(1)}, F_1^{(1)}, F_2^{(1)}, B_2^{(1)}, B_1^{(1)}, B_0^{(1)}\right),$$

so a single model learns the forward and backward computations jointly.

**Data Generation.**   For each seed, we sample a teacher 3-layer FFN whose weight entries satisfy

$$(W_i)_{ab} \sim \mathcal{N}\left(0, \frac{1}{d}\right),$$

together with an independent student initialization. For each input $x \sim \mathcal{N}(0, I)$, we generate $y = f_{\text{teacher}}(x)$. and then compute the exact one-round Section 4 trajectory from the student initialization and the pair $(x, y)$. Thus each training example consists of a prompt together with its exact six-block continuation.

**Training Objective.**   We train autoregressively on the exact continuation using masked MSE over the trajectory blocks

$$F_0^{(1)}, F_1^{(1)}, F_2^{(1)}, B_2^{(1)}, B_1^{(1)}, B_0^{(1)}.$$

Equivalently, the loss is computed only on the forward/backward trajectory tokens, not on the prompt tokens.

**Evaluation and Metrics.**   We evaluate in two modes. In teacher-forced evaluation, each block is predicted conditioned on the true previous blocks. In free-running evaluation, after the prompt the model conditions on its own generated blocks. For each block

$$F_0, \ F_1, \ F_2, \ B_2, \ B_1, \ B_0,$$

we report its token MSE, and we also report the overall MSE averaged across all six blocks. All values are averaged over 10 random seeds.

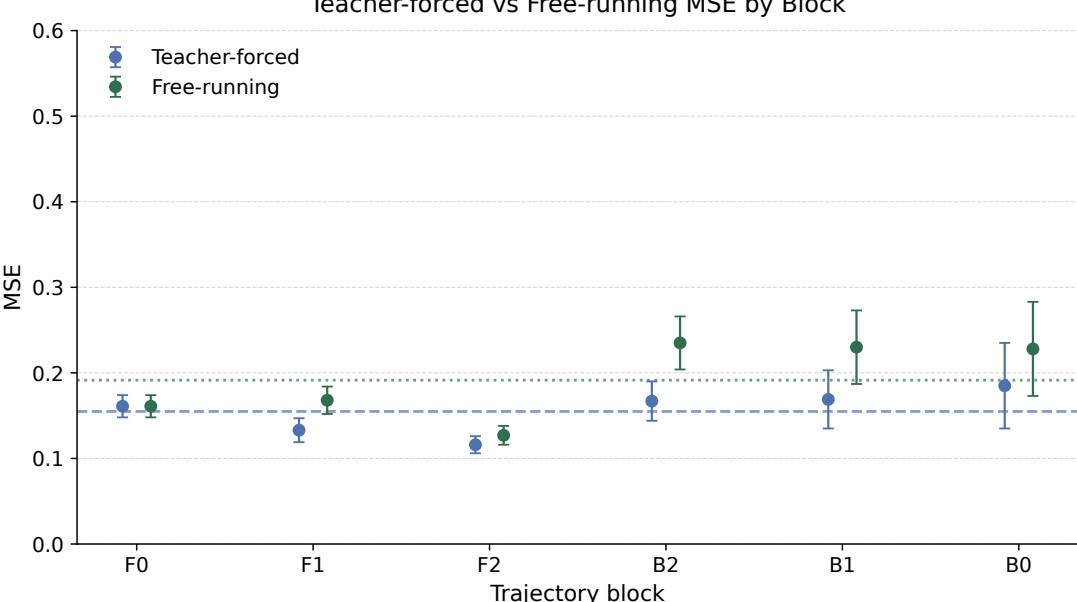

*Figure 4.* **End-to-End Training of ICGD of FFN.** We sample a 3-layer teacher FFN and a separate student FFN for initialization. For each input $x \in \mathbb{R}^d \sim N(0, I)$, we generate $y$ from a teacher FFN. The transformer receives the input $[x, y, \text{initial layer states}]$ and trained to autoregressively predicts the full trajectory $(F_0, F_1, F_2, B_2, B_1, B_0)$. Teacher-forced uses the true previous tokens to make a prediction. Free-running MSE uses the model's own previous outputs to predict the next token. The figure reports per-block MSE averaged over 10 seeds; the overall means are $0.155$ and $0.191$, respectively. The small gap shows that one shared transformer learns the forward and backward updates jointly.

**Results.** Across 10 seeds, the overall teacher-forced MSE is $0.155$, while the overall free-running MSE is $0.191$. The gap remains small across both forward and backward blocks. This shows that one shared transformer learns the full one-round ICGD trajectory jointly, rather than relying on separate forward and backward models.

Finally, we test whether the same trained transformer can be reused recursively beyond the single-round setting. We perform a 10-round rollout and track the resulting student-weight recovery error. As shown in Figure 4, the error grows gradually under repeated reuse, similar to our theoretical results from Proposition 4.4, although remaining milder than the worst-case accumulation bound.

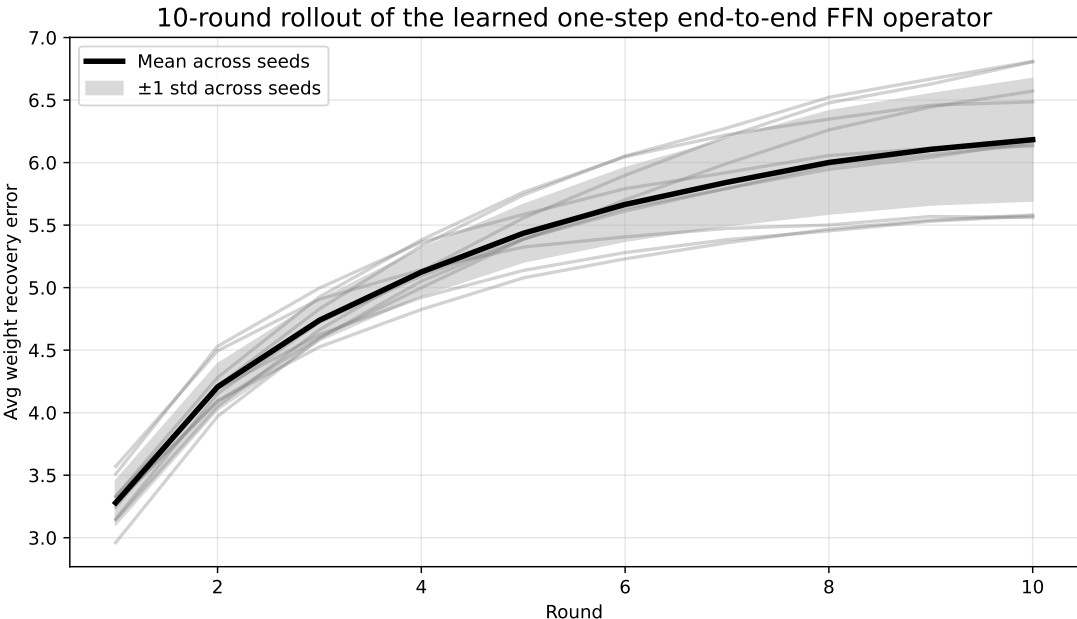

*Figure 5.* **10-Round Rollout of ICGD of FFN.** For each of 10 random seeds, we first train the one-step end-to-end transformer in Figure 4 . We then reuse the same transformer for 10 rounds on fresh $(x_l, y_l)$ pairs. At the beginning of each round, the input is rebuilt from the current $(x_l, y_l)$ and the predicted updated weights from the previous round. The transformer then autoregressively predicts the local trajectory $(F_0, F_1, F_2, B_2, B_1, B_0)$ for that round. The figure reports the weight recovery error at each round, averaged over seeds; the black curve is the mean and the shaded region is $\pm 1$ standard deviation. The error increases under repeated reuse, consistent with Proposition 4.4, although not the worst-case linear trend.

