# OpenReview forum: "Chain-of-Thought Gradient Descent"
_ICML.cc/2026/Conference — ICML 2026 regular_

### Official Review · Reviewer_E4ra · 2026-03-11

**Soundness:** 3
**Presentation:** 3
**Significance:** 2
**Originality:** 2
**Overall Recommendation:** 4
**Confidence:** 2

**Summary:**

This paper studies how transformers with Chain-of-Thought (CoT) prompting can simulate multi-step optimization procedures in context. The authors propose a theoretical framework where a single-layer transformer recursively generates a CoT trajectory that mimics the forward and backward passes of gradient descent. A key idea is dynamic masking, which restricts attention to only the tokens relevant for the current computation step, enabling reuse of the same transformer block while avoiding redundant processing. Using this mechanism, the authors construct transformers that approximate in-context gradient descent for several statistical models and show how to simulate gradient descent for deep neural networks with improved computational efficiency.

**Compliance With Llm Reviewing Policy:**

Affirmed.

**Final Justification:**

During the rebuttal, the authors clarified the motivation behind the results in Section 3 and provided details on the size of the Transformer models in terms of parameter count. They also discussed the rationale for dynamic masking and context windows. Based on these clarifications, I will raise my score.

**Key Questions For Authors:**

See weaknesses.

**Limitations:**

Yes.

**Strengths And Weaknesses:**

**Strengths:**

* The paper is generally well written, and the main ideas are presented in a clear and easy-to-follow manner.
* The attempt to connect chain-of-thought with ICGD is interesting and potentially valuable.

**Weaknesses:**

**Major:**

* The paper does not provide a clear definition or specification of the Transformer architecture used in the analysis.
* In the theoretical results, the authors do not specify how large the Transformer model needs to be in terms of the number of heads, head dimension, embedding dimension, or overall number of parameters. This makes it difficult to understand the exact regime in which the theorems apply.
* The results in Section 3 appear somewhat incremental, as they are largely built upon prior work by Bai et al. (2023) and Hu et al. (2025).
* Several simplifying techniques are introduced to facilitate the theoretical analysis, such as dynamic masking and a hand-crafted context window that forces the model to focus only on the last $n$ tokens. However, these assumptions are not realistic in practical Transformer implementations. Due to the nature of the Softmax attention mechanism, tokens typically attend to all other tokens. The use of dynamic masking appears more like a workaround to avoid the difficulty of dealing with the Softmax attention mechanism. While causal masking is widely used and has a clear functional role, the practical motivation or meaning of dynamic masking is not convicing. Relying on such artificial mechanisms may weaken the contribution and potentially oversimplify the problem.

**Minor:**

* Typo: In Section 3, “lass” should be “lasso.”

---

> ### Author Rebuttal · Authors · 2026-03-29
>
> We thank the reviewer for the detailed feedback. We address each weakness below.
>
> ---
>
> > ### W1. The paper does not provide a clear definition or specification of the Transformer architecture used in the analysis.
>
> Thanks for pointing this out. We agree we should be specific on the transformer architecture considering many variants in the literature. Our transformer uses the same definition of softmax attention and ReLU FFN as in [1]. Due to the page limits, we omit it in the first version of submission. We will include the transformer definition in the final version.
>
> ---
>
> > ### W2. In the theoretical results, the authors do not specify how large the Transformer model needs to be in terms of the number of heads, head dimension, embedding dimension, or overall number of parameters. This makes it difficult to understand the exact regime in which the theorems apply.
>
> Thank you for raising this concern. We clarify the model complexity in two points.
>
> * For section 3, the construction is based on lemma 3.1. We follow [1] section E.2 derivation. Fix a one-step ICGD error $\epsilon$. We have $H_{\mathrm{total}}=\Theta(d \cdot H_{\mathrm{relu}}(\epsilon)\cdot \frac{1}{(n-2)\epsilon})$.  Here $d$ is input dimension, $n$ is number of in-context examples. $H_{\mathrm{relu}}$ is the  approximation error of loss gradient by ReLU NN ([1] Eq. (E.66)). In grid-based approximation, $H_{\mathrm{relu}}$ depends on the modulus of continuity of the target function. For squared loss (ridge), this is $O(1)$. For standard smooth GLM losses, $H_{\mathrm{relu}} = O(1/\epsilon^2)$. For lasso loss, we only adds an FFN block to approximate proximal operator, hence attention head scale as squared loss case. The total parameters scale with $d^2 H_{\mathrm{total}}$.
>
> * For section 4 construction, architecture size is a polynomial in $d$ and does not grow with simulated depth $N$ or the number of GD rounds $L$. Specifically, the forward/backward constructions use $O(d)$ heads. Hidden dimension for attention and FFN are $O(d)$, hence head parameters dominate and scale in $O(d^3)$. Proposition 4.3 combines them through a fixed router. This changes the architectural size by only a constant factor.
>
> [1] “Universal Approximation with Softmax Attention” 2025
>
> ---
>
> > ### W4. The results in Section 3 appear somewhat incremental, as they are largely built upon prior work by Bai et al. (2023) and Hu et al. (2025).
>
> We thank the reviewer for the comments, and clarify as follows.
>
> **The role of Section 3 is to provide intuition under simplified settings.** It’s simple, but necessary. Without Section 3, the masking mechanism in Section 4 would be harder to grasp.
>
> This setting makes two points explicit:
> * CoT realizes multi-step updates without increasing transformer depth (contrary to Bai’s work)
> * Bounded context access controls per-step computation (simplest example of dynamic masking)
>
> Beyond this, we also emphasize the importance of studying the statistical models in section 3. See our response to **reviewer ivTB, W3** for detailed discussions.
>
> ---
>
> > ### W4. Several simplifying techniques are introduced to facilitate the theoretical analysis, such as dynamic masking and a hand-crafted context window that forces the model to focus only on the last n tokens. However, these assumptions are not realistic in practical Transformer implementations .... Relying on such artificial mechanisms may weaken the contribution and potentially oversimplify the problem.
>
>
> Thanks for bringing this up. We clarify 2 points:
>
> * Dynamic masking is the formalized tool to demonstrate the CoT-only efficiency guarantee. Our result first identifies CoT advantages over fixed weight Transformer's forward pass in ICL. This doesn’t rely on dynamic masking. See our response to **Reviewer ivTB, W1/W2**.
>
> * Built on top of this, we also show an efficiency guarantee only available in CoT settings. Expandable workspace enables the model to only process relevant information in-context. Dynamic masking is the way we formalize this idea. Altogether we show provable usefulness and efficiency of the CoT mechanism.
>
> * The practical motivation is when generating CoT trajectory, it is not efficient to attend to all the tokens when sequences become long. One should only attend to more recent token, such as slicing window attention [1-3]. Recent work also shows CoT benefits by dynamically only processing the relevant token in CoT trajectory [4]. We hence believe our theorems are close to practical Transformer implementations.
>
> [1] "Gemma 2: Improving open language models at a practical size." (2024)
>
> [2] "Olmo 3." (2025)
>
> [3] "Mistral 7b” (2023)
>
> [4] "Pencil: Long thoughts with short memory." ICML (2025)
>
> ---
>
> Thank you again for the time and effort invested in our paper. We have addressed all your concerns and questions with utmost care. Please let us know if there is anything we can clarify further.

---

> > ### Author Rebuttal · Reviewer_E4ra · 2026-04-03
> >
> > Thanks for the response. I will raise my score to 4.

---

> > > ### Author Response · Authors · 2026-04-03
> > >
> > > Thanks for the consideration! We are very happy to see our rebuttal meets your expectations. We will incorporate your comments and clarified points in our final version. Your constructive feedback certainly helps us improve this work!

---

### Official Review · Reviewer_JpXP · 2026-03-13

**Soundness:** 3
**Presentation:** 2
**Significance:** 2
**Originality:** 3
**Overall Recommendation:** 4
**Confidence:** 3

**Summary:**

The paper shows that Chain-of-Thought (CoT) enables a single-layer transformer to approximate the training process of a multi-layer feed-forward network in-context, demonstrating the expressive power of CoT. In addition, it proposes an approach named dynamic masking to reduce the total computation compared with prior works.

**Compliance With Llm Reviewing Policy:**

Affirmed.

**Final Justification:**

The rebuttal has addressed my concerns. I will raise my score to 4.

**Key Questions For Authors:**

See Weaknesses.

**Limitations:**

The authors haven't discussed the limitations.

**Strengths And Weaknesses:**

Strengths:
1. The paper applies CoT and dynamic masking to the transformer model to simulate gradient descent for FFN. It provides a solid theoretical foundation for how transformers can simulate complex optimization algorithms.
2. By introducing a novel dynamic masking mechanism, the model only needs to focus on the current step, which reduces the computation.

Weaknesses:
1. In definition 4.3, the bottom rows of $X_l$ and $Y_l$ are $2_{1\times d}$, which seems unusual. The authors could discuss the motivation for this design and show the necessity of this part.
2. The paper proposes dynamic masking to focus on two current tokens within each update. However, computing the mask itself requires pre-knowledge or additional training. In more complex settings, it remains unclear how the model can determine which blocks to attend to.
3. The theoretical proofs are mainly for the ReLU activation function, while in practice, other activations like GeLU and SwiGLU are also widely used. It's better to discuss more complex functions.
4. The experimental results are limited to a relatively small-scale setting, and the forward and backward operators are trained separately, which differs from the end-to-end learning of standard transformers. In addition, the experiments corresponding to statistical models in Section 3 would help strengthen the empirical evidence.

---

> ### Author Rebuttal · Authors · 2026-03-31
>
> > ### W1. In definition 4.3, the bottom rows of  $X_l$ and $Y_l$ are $2_{1 \times d}$, ... discuss the motivation for this design and show the necessity of this part.
>
> Thanks for the comment. To clarify, **the bottom rows in $X_l$ and $Y_l$ are tags to help identify the input.** Such tag is a common setup ICL theory. For example, people use last row of input as tag for training/testing ICL example [1].
>
> Concretely, they are routing tags. Their role is to let the single update block in Proposition 4.3 distinguish the four routed input, and then apply the correct sub computation. So the necessity is only for the routing mechanism, not for the in-context simulated NN itself.
>
> However, we agree that this should be stated more explicitly. In the final revision, we will add a footnote in Definition 4.3 to clarify. These bottom rows are used only for routing, and that other distinct constants would also work.
>
>
> ---
>
> > ### W2. The paper proposes dynamic masking ... requires pre-knowledge or additional training...
>
> Thanks for bringing this up. We agree that our current setup is still gapped from the general and practical scenarios (with learning included). We deem such directions exciting future works.  However,  we want to remind the reviewer our scope is to provide provable advantages of CoT.  To achieve this goal with high rigor, it’s necessary to consider specialized settings like we did. We believe as long as the considered setups capture important essences of practice, such treatment is acceptable.
>
> In particular, our setup captures the essence of only process relevant tokens in a long CoT trajectory.
>
> ---
>
> > ### W3. The theoretical proofs are mainly for the ReLU activation function, while in practice, other activations like GeLU and SwiGLU are also widely used.
>
> Thanks for the suggestion. The extension to these two activations is trivial, since they are approximable by sum of ReLU functions. We explain the concrete reasons below.
>
> Two part in our paper have ReLU FFN:
> * **Target network simulated in-context (Section 4).** Our goal is to certify that a frozen transformer with CoT simulates nontrivial learning dynamics in-context. ReLU already captures full expressiveness, since ReLU networks are universal approximators. Replacing ReLU by smooth activations does not change expressiveness. Such choice is also common in prior works [1,2]
>
> * **FFN in the constructed transformer.** Our construction does not rely on ReLU as a special function. On any bounded domain, smooth activations such as GeLU and SwiGLU admit uniform piecewise-linear approximations, and thus admit ReLU network approximations to error $\epsilon$. Our error-propagation bound then applies with adjusted constants. We will add a remark formalizing this reduction.
>
> ---
>
> > ### W4. The experimental results are limited to a relatively small-scale setting, and the forward and backward operators are trained separately ... the experiments corresponding to statistical models in Section 3 would help ...
>
> Thanks for pointing this out. We further conducted 3 more experiments. The detailed setting is in the caption of figure.
>
> * **End-to-End Training of ICGD of FFN.** We sample a teacher $3$-layer FFN, generate one exact GD round trajectory on synthetic $(x,y)$ examples, and train a single transformer to autoregressively predicts the full trajectory $[F^{(1)}_0,,...B^{(1)}_0]$. Hence the forward and backward updates are learned jointly. Over 10 random seeds, we obtain consistent results : teacher-forced (MSE = 0.155) is close to free running (MSE = 0.191) .  This demonstrate model successfully learn to perform proposition 4.3, instead of train separate two transformers to compute forward and backward block. See the link for the result: https://imgur.com/a/pqghOjy.
>
> * **10-Round Rollout of ICGD of FFN.** We also add a CoT rollout experiment for the above trained model to simulate FFN dynamics. Across 10 seeds, the mean weight recovery error grows from Round 1 to Round 10, consistent with Proposition 4.4, although not the worst-case linear trend. See the link for the result: https://imgur.com/a/pYB7Eki.
>
> * **ICGD of Statistical Models.** Finally, we add statistical model experiments for GLM, ridge, and lasso. We generate synthetic (x,y) examples, compute the exact one step target update from each model’s ground truth optimization rule, and train a single shared transformer to autoregressively predict the one-step GD update. Across 10 random seeds, free running MSE remains very close to teacher forced MSE for all three models. See the link for the result: https://imgur.com/a/Dcqfz8v.
>
> ---
>
> [1] "Transformers as statisticians: Provable in-context learning with in-context algorithm selection." NeurIPS (2023)
>
> [2] "In-Context Deep Learning via Transformer Models." ICML (2025)
>
> ---
>
> Thank you again for the valuable feedback. We hope our clarification addresses your concerns and welcome further discussion!

---

> > ### Author Rebuttal · Reviewer_JpXP · 2026-04-04
> >
> > Thank you for your response. I think my concerns have been addressed. I will raise my score to 4.

---

> > > ### Author Response · Authors · 2026-04-04
> > >
> > > Thank you again for your constructive comments! They are very helpful. Happy to see that our rebuttal is satisfactory! Wish you a wonderful day!

---

### Official Review · Reviewer_ivTB · 2026-03-19

**Soundness:** 4
**Presentation:** 4
**Significance:** 2
**Originality:** 2
**Overall Recommendation:** 5
**Confidence:** 3

**Summary:**

The paper improves previous constuction show that Chain-of-Thought can let a single-layer transformer simulate multi-step in-context gradient descent and backpropagation for an N-layer feed-forward network. They do this in a a way that reduces compute by O(N). They also extends the analysis to GLM, ridge regression, and lasso.

**Compliance With Llm Reviewing Policy:**

Affirmed.

**Final Justification:**

As noted in my rebuttal acknowledgment, the authors’ response convinced me that the paper’s efficiency claims are meaningful and address one of my main concerns about the contribution. As a result, my evaluation became more positive. Overall, I believe this paper makes a valuable contribution and would be a good addition to the ICML program.

**Key Questions For Authors:**

I realize this is partly a matter of personal taste, but could you explain why, in your view, finding a more efficient construction is important in this setting? Relatedly, could you also try to convince me that the constructions for linear models, ridge regression, and lasso regression are important for reasons beyond providing intuition for the neural-network part?

I know this is ultimately a matter of taste, and so it is unlikely that I will fully change my mind. That said, if I were convinced that either of these two aspects is sufficiently important, I would be willing to raise my score.

**Limitations:**

yes

**Strengths And Weaknesses:**

- The paper is well written, tackles an interesting problem, and I enjoyed reading it.

- My main reservation is that the paper improves the efficiency of a construction in a setting where, to me, the more interesting question is whether this phenomenon is reasonably possible at all, rather than how to obtain the most efficient construction once it is already known to be possible. The main gain appears to be an \(O(N)\) improvement, and while the masking-based implementation is nice, I find this contribution somewhat incremental. Relatedly, it does not feel like the previous results were so prohibitively large or inefficient that this improvement substantially changes the overall picture, so I am unsure whether the contribution is sufficient.

- That said, despite the above concerns, I do think this work has a place in the ML community and could clear the bar in some settings. I am just unsure whether it clearly meets the NeurIPS bar, so I would leave that final judgment to the AC.

- I find the neural network part of the paper substantially more interesting than the linear-model, ridge-regression, and lasso-regression parts, and therefore my review is based mostly on the NN analysis.

---

> ### Author Rebuttal · Authors · 2026-03-29
>
> > ### W1. My main reservation is that the paper improves the efficiency of a construction in a setting where, to me, the more interesting question is whether this phenomenon is reasonably possible at all, rather than how to obtain the most efficient construction once it is already known to be possible. The main gain appears to be an (O(N)) improvement, and while the masking-based implementation is nice, I find this contribution somewhat incremental. Relatedly, it does not feel like the previous results were so prohibitively large or inefficient that this improvement substantially changes the overall picture, so I am unsure whether the contribution is sufficient.
> > ### W2. That said, despite the above concerns, I do think this work has a place in the ML community and could clear the bar in some settings. I am just unsure whether it clearly meets the NeurIPS bar, so I would leave that final judgment to the AC.
>
> Thanks for careful reading and raising this concern. We clarify 3 points:
>
> **First, we don't just rely on masking to achieve efficient results.** Our analysis provides sharp distinctions between a fixed weight Transformer's forward pass and CoT under ICL setup:
>
> - We consider "ICGD of deep learning model (ReLU NNs)" task as it captures key essences of ICL (fixed-weight model learns from context through forwardpass)
>
> - A transformer model with fixed depth/hidden dimension has fixed capacity for ICGD. Consequently, it requires larger the hidden dimension/depth to simulate larger models/performing more step ICGD.
>
> - By contrast, CoT provides an expandable workspace (the trajectory). It doesn't require a deeper model/larger hidden dimension for more step ICGD/larger simulated models.
>
> In this sense, CoT enables arbitrary step/capacity learning dynamics (non-convex optimization on NN) within constant transformer depth. Hence our results provide strong theoretical guarantees of the CoT mechanism.
>
> **Second, we also show efficiency guarantee only available in CoT setting.** Expandable capacity + dynamic masking avoids the redundant ``process everything'' cost of a deep transformer without CoT.
>
>
> **Lastly, as of rationale of dynamic masking, it is in fact mimicing the practical component in LLM.** For example, dynamic masking in Sec 3 is related to sliding window attention used in LLM [1-3], and sparse attention in LongFormer [4]. Recent work also shows CoT benefits by dynamically only processing the relevant token in CoT trajectory [5].
>
> [1] "Gemma 2: Improving open language models at a practical size." (2024)
>
> [2] "Olmo 3." (2025)
>
> [3] "Mistral 7b” (2023)
>
> [4] Beltagy, Iz, Matthew E. Peters, and Arman Cohan. "Longformer: The long-document transformer." (2020)
>
> [5] Yang, Chenxiao, et al. "Pencil: Long thoughts with short memory." ICML (2025)
>
> ---
>
> > ### W3: I find the neural network part of the paper substantially more interesting than the linear-model, ridge-regression, and lasso-regression parts, and therefore my review is based mostly on the NN analysis. Relatedly, could you also try to convince me that the constructions for linear models, ridge regression, and lasso regression are important for reasons beyond providing intuition for the neural-network part?
>
> Thanks for your kind words and questions.
>
> **Why GLM/Ridge/Lasso Matter?** Yes you’re correct that the main purpose of Sec 3 is indeed to provide intuition for Sec 4. Beyond this, it allows our work to be well positioned in literature [1-5]. Lastly, it proves an optimization guarantee on canonical convex objectives. It bounds the distance (or objective gap) to the true optimizer $w*$ after L CoT steps. Sec 4 does not give this type of guarantee because deep-network training is nonconvex.
>
> [1] "What can transformers learn in-context? a case study of simple function classes." NeurIPS (2022)
>
> [2] "What learning algorithm is in-context learning? investigations with linear models." ICLR (2023)
>
> [3] "Transformers as algorithms: Generalization and stability in in-context learning." ICML (2023)
>
> [4] "Transformers learn in-context by gradient descent." ICML (2023)
>
> [5] "Transformers as statisticians: Provable in-context learning with in-context algorithm selection." NeurIPS (2023)
>
> ---
>
> > ### Q1: I realize this is partly a matter of personal taste, but could you explain why, in your view, finding a more efficient construction is important in this setting?
>
> In response, **this paper aims to provide a theoretical advantage/analysis of the CoT.** Its recursive nature + multistep reasoning provides a unique mechanism to process information in a cleverer way compared to single-pass deep models. **The efficiency construction is a mathematical way to formalize these differences.**
>
> Second, **our efficiency guarantees matter in real-world CoT.** Long CoT faces a memory bottleneck. Intermediate computations accumulate in context after they become irrelevant. CoT + dynamic masking enables only process relevant tokens.

---

> > ### Author Rebuttal · Reviewer_ivTB · 2026-04-03
> >
> > The answer to Q1 addressed my main concern and convinced me to update my score to Accept.
> >
> > I still encourage the authors to reorganize the paper and place less emphasis on the linear-model, ridge-regression, and lasso-regression parts. That said, I now believe the paper merits acceptance.

---

> > > ### Author Response · Authors · 2026-04-03
> > >
> > > Amazing! Very glad your concerns are addressed. Also thank you for score raising! We will tone down that part and reposition more it as intuitive examples. We will also revise the draft according to our discussions. Wish you a wonderful day!

---

### Decision · Program_Chairs · 2026-04-30

**Decision:**

Accept (regular)

**Comment:**

The paper makes a solid theoretical contribution to understanding CoT by showing how a single-layer transformer can simulate multi-step in-context gradient descent with improved efficiency via dynamic masking. Reviewers found the core ideas interesting and technically sound, and the rebuttal successfully addressed concerns about significance, architectural specification, masking assumptions, and empirical scope; all active reviewers raised or maintained accept-level scores. While the practical realism of some assumptions and the presentation of Section 3 can still be improved, I believe the theoretical novelty, clear analysis, and strong rebuttal justify acceptance.